# SCORE DISTILLATION OF FLOW MATCHING MODELS

## ABSTRACT

Diffusion models achieve high-quality image generation but are limited by slow iterative sampling. Distillation methods alleviate this by enabling one- or few-step generation. Flow matching, originally introduced as a distinct framework, has since been shown to be theoretically equivalent to diffusion under Gaussian assumptions, raising the question of whether distillation techniques such as score distillation transfer directly. We provide a simple derivation—based on Bayes' rule and conditional expectations—that unifies Gaussian diffusion and flow matching without relying on ODE/SDE formulations. Building on this view, we extend Score identity Distillation (SiD) to pretrained text-to-image flow-matching models, including SANA, SD3-MEDIUM, SD3.5-MEDIUM/LARGE, and FLUX.1-DEV, all with DiT backbones. Experiments show that, with only modest flow-matching- and DiT-specific adjustments, SiD works out of the box across these models, in both data-free and data-guided settings, without requiring teacher finetuning or architectural changes. This provides the first systematic evidence that score distillation applies broadly to text-to-image flow matching models, resolving prior concerns about stability and soundness and unifying acceleration techniques across diffusion- and flow-based generators.

## 1 INTRODUCTION

Diffusion models (Sohl-Dickstein et al., 2015; Song & Ermon, 2019) have achieved remarkable image generation quality, but their slow inference speed remains a longstanding challenge, as sampling requires solving an SDE or ODE through iterative refinement. Early models required hundreds or even thousands of steps (Ho et al., 2020b; Song et al., 2021), though recent work has accelerated generation by improving samplers for pretrained models (Song et al., 2020; Lu et al., 2022; Liu et al., 2022a; Karras et al., 2022) or distilling them into one- or few-step generators (Luhman & Luhman, 2021; Zheng et al., 2022; Salimans & Ho, 2022; Luo et al., 2023b; Yin et al., 2024b; Zhou et al., 2024). Flow matching was later introduced as an alternative framework, motivated by the hope that straighter ODE trajectories would require fewer integration steps—most notably in rectified flow (Liu et al., 2022b; Lipman et al., 2022). Although initially formulated with different objectives, rectified flow has since been shown theoretically interchangeable with diffusion models under Gaussian assumptions (Kingma & Gao, 2023; Ma et al., 2024; Gao et al., 2024). Nevertheless, practical differences remain, including variations in noise schedules, loss weighting, and architectures.

This theoretical equivalence raises a natural question: can diffusion distillation techniques—broadly divided into trajectory and score distillation (Fan et al., 2025), and proven effective for compressing pretrained diffusion models into one- or few-step generators—be directly applied to flow-matching models? Prior work has begun to explore this. The continuous-time consistency model (Lu & Song, 2024) introduced TrigFlow and demonstrated trajectory distillation for pretrained TrigFlow models. Extending this to text-to-image (T2I) generation, Chen et al. (2025) developed SANA-Sprint by reformulating SANA (Xie et al., 2024) from rectified flow into TrigFlow and applying consistency distillation. While effective, this approach requires nontrivial finetuning of rectified-flow checkpoints into TrigFlow counterparts, making it inapplicable to pretrained rectified-flow models without additional adaptation.

Score distillation relaxes the constraint of strictly following the teacher's sampling trajectory and has shown consistent gains over trajectory-based consistency distillation on diffusion benchmarks such as CIFAR-10 and ImageNet (Zhou et al., 2025c). Yet its applicability to flow-matching T2I models remains unclear. If effective, a further question is whether additional adaptation steps—such as

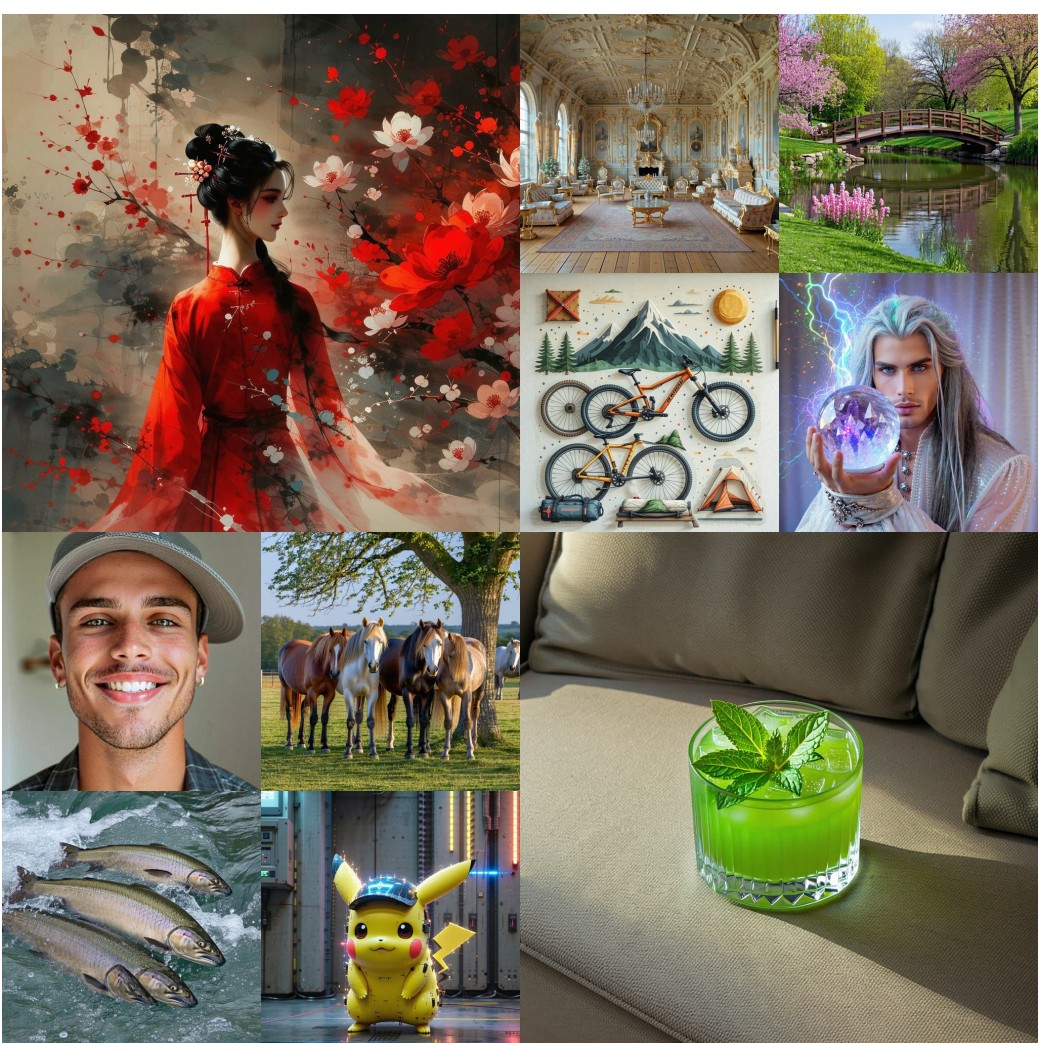

Figure 1: Qualitative results produced by the four-step SiD-DiT generator distilled from SD3.5-LARGE.

finetuning, as in SANA-Sprint—are necessary. This uncertainty is compounded by a sensitive design space, including noise schedules, loss weighting, network preconditioning (Karras et al., 2022), and architecture. Small changes in these factors can significantly affect performance, as evidenced by methods like sCM, which require careful adaptation during pretraining (Lu & Song, 2024) or finetuning (Chen et al., 2025). Concerns about stability further complicate matters: consistency distillation was favored in SANA-Sprint partly due to instability observed in Distribution Matching Distillation (DMD) (Yin et al., 2024c;a). However, it remains unclear whether this instability is unique to DMD's KL-based formulation or reflects broader issues in score distillation, which can also be defined with divergences such as Fisher divergence (Zhou et al., 2024) or $f$-divergences (Xu et al., 2025). Huang et al. (2024) argue flow matching does not explicitly model probability density, raising doubts about the soundness of applying distribution-divergence-based objectives directly.

In this work, we revisit these questions and clarify common misconceptions surrounding diffusion and flow matching. We present a unified perspective showing that, under Gaussian assumptions, their optimal solutions are theoretically equivalent, differing primarily in the weight-normalized distribution of time steps. Our derivation avoids ODE/SDE formulations and instead relies on Bayes' rule, conditional expectations, and properties of the squared Euclidean distance to reconcile diverse loss functions. This analysis underscores the equivalence of diffusion and flow-matching objectives while also highlighting practical differences in weighting, scheduling, and architectural design.

To validate this view, we adopt the few-step Score identity Distillation (SiD) framework (Zhou et al., 2025a), previously shown effective for diffusion models such as SD1.5 and SDXL with U-Net

backbones. Here, we extend SiD to pretrained flow-matching models with Diffusion Transformer (DiT) (Peebles & Xie, 2023) backbones, including SANA (Xie et al., 2024; Chen et al., 2025), SD3-MEDIUM, SD3.5-MEDIUM, SD3.5-LARGE, and FLUX.1-DEV (Labs, 2024), spanning 0.6B–12B parameters (2.4–48 GB in fp32). We show that SiD works out of the box across these models in both data-free and data-guided settings: the former requires no additional images beyond the teacher, while the latter incorporates adversarial learning by pooling discriminator features along the spatial dimension from a suitable DiT layer without introducing new parameters.

We provide a review of related work in Appendix B. **Our code is provided in the Supplementary Material.** Importantly, a single codebase and hyperparameter configuration suffice across all T2I flow-matching models, underscoring the robustness and applicability of the SiD-DiT framework.

## 2 A UNIFIED VIEW OF DIFFUSION AND FLOW MATCHING

The pretraining objective of a diffusion model can be framed as predicting different targets—such as the score function, the clean image $x_0$, the noise $\epsilon$, or the velocity—all of which are theoretically equivalent under certain assumptions and perspectives (Albergo et al., 2023; Kingma & Gao, 2023; Ma et al., 2024; Gao et al., 2024; Geffner et al., 2025). We make these equivalences explicit by conditioning on the noisy observation $x_t$. Given the conditional expectation of one target (e.g., $\mathbb{E}[x_0 \,|\, x_t]$), the others (e.g., $\mathbb{E}[\epsilon \,|\, x_t]$) follow through linear transformations. The key distinction across these formulations lies in the weighting of timesteps within the training loss, which drives differences in learning dynamics and empirical performance despite their shared structure.

### 2.1 TWEEDIE'S FORMULA IN DIFFUSION AND FLOW-MATCHING MODELS

We deliberately avoid the standard SDE/ODE formulation, unnecessary for score distillation. This simplifies the discussion and lets us focus on training losses, independent of their motivations or parameterizations. Specifically, we rewrite both diffusion and flow matching losses as expectations under $p(x_0 \,|\, x_t)$, the conditional distribution of the clean image $x_0$ given the corrupted one $x_t$, and then apply Tweedie's formula together with a standard identity for the squared Euclidean distance.

All Gaussian-based diffusion and flow matching models corrupt the data according to

$$x_t = \alpha_t x_0 + \sigma_t \epsilon, \quad x_0 \sim p_{\text{data}}(x_0), \quad \epsilon \sim \mathcal{N}(0, I), \tag{1}$$

where $\alpha_t, \sigma_t > 0$, and the signal-to-noise ratio (SNR), defined as $\text{SNR}_t = \frac{\alpha_t^2}{\sigma_t^2}$, decreases monotonically from infinity to zero as $t$ increases from zero to its maximum value (*e.g.*, 1 for continuous time or $T = 1000$ for discrete time). Despite the varied parameterizations of $\alpha_t$ and $\sigma_t$—such as $\alpha_t^2 + \sigma_t^2 = 1$ in variance-preserving diffusion and TrigFlow, or $\alpha_t + \sigma_t = 1$ in rectified flow—all formulations can be reconciled by aligning their implied $\text{SNR}_t$ trajectories over the diffusion process, up to scaling differences. These scaling factors can be absorbed into the preconditioning of the underlying neural networks (Karras et al., 2022).

In Gaussian diffusion, the marginal distribution of the forward-diffused variable $x_t$ is given by

$$p(x_t) = \int q(x_t \,|\, x_0)\, p_{\text{data}}(x_0)\, \mathrm{d}x_0, \quad q(x_t \,|\, x_0) = \mathcal{N}(x_t; \alpha_t x_0, \sigma_t^2). \tag{2}$$

The conditional distribution of the clean data $x_0$ given the noisy observation $x_t$ can be written as

$$p(x_0 \,|\, x_t) = \frac{q(x_t \,|\, x_0)\, p_{\text{data}}(x_0)}{p(x_t)}, \tag{3}$$

which follows directly from Bayes' rule, and the conditional expectation of $x_0$ given $x_t$ is given by

$$\mathbb{E}[x_0 \,|\, x_t] = \int x_0\, p(x_0 \,|\, x_t)\, \mathrm{d}x_0. \tag{4}$$

A key property of Gaussian diffusion is that the score of the marginal distribution $p(x_t)$, given by $\nabla_{x_t} \log p(x_t)$, is related to the conditional expectation $\mathbb{E}[x_0 \,|\, x_t]$ as

$$\nabla_{x_t} \log p(x_t) = -\frac{x_t - \alpha_t\, \mathbb{E}[x_0 \,|\, x_t]}{\sigma_t^2}.$$

This identity, known as Tweedie's formula (Robbins, 2020; Efron, 2011; Chung et al., 2022), can be derived by interchanging differentiation and integration in (2), using the fact that the score of Gaussian is analytic: $\nabla_{x_t} \ln q(x_t \,|\, x_0) = -\frac{x_t - \alpha_t x_0}{\sigma_t^2}$, and applying Bayes' rule in (3) and conditional expectation in (4). Therefore, the score estimation problem is equivalent to estimating $\mathbb{E}[x_0 \,|\, x_t]$.

## 2.2 Equivalence of Diffusion and Flow-Matching Objectives and Variants

**Diffusion with $x_0$-Prediction.** Estimating the true $x_0$ given $x_t$ is often called $x_0$-prediction, though a more precise term is $x_0$-mean-prediction: the mapping from $x_t$ to $x_0$ is one-to-many, and the best one can do is to recover the conditional mean of all possible $x_0$ values that could have produced $x_t$ under the forward diffusion process. The corresponding loss used in diffusion to serve this purpose is

$$L_\phi(x_t) = \mathbb{E}_{x_0 \sim p(x_0 \mid x_t)} \left[ \|f_\phi(x_t, t) - x_0\|_2^2 \right]. \tag{5}$$

To estimate $\mathbb{E}_{x_t \sim p(x_t)}[L_\phi(x_t)]$, we draw $(x_0, x_t)$ in practice not from $p(x_0 \mid x_t)\, p(x_t)$, but from $q(x_t \mid x_0)\, p_{\text{data}}(x_0)$, which defines the same joint distribution and is straightforward to sample from.

One can show that the optimal solution to the above loss is

$$f_{\phi^*}(x_t, t) = \mathbb{E}[x_0 \mid x_t]. \tag{6}$$

This can be established in two ways. One approach is to observe that the squared Euclidean distance is a Bregman divergence and apply Lemma 1 from Banerjee et al. (2005); see also Zhou et al. (2023) for a more detailed discussion from this perspective. Another approach is to decompose this loss as:

$$L_\phi(x_t) = \mathbb{E}_{x_0 \sim p(x_0 \mid x_t)} \left[ \|(f_\phi(x_t, t) - \mathbb{E}[x_0 \mid x_t]) - (x_0 - \mathbb{E}[x_0 \mid x_t])\|_2^2 \right]$$

$$= \mathbb{E}_{x_0 \sim p(x_0 \mid x_t)} \left[ \|f_\phi(x_t, t) - \mathbb{E}[x_0 \mid x_t]\|_2^2 \right] + C,$$

where $C = \mathbb{E}_{x_0 \sim p(x_0 \mid x_t)} \left[ \|x_0 - \mathbb{E}[x_0 \mid x_t]\|_2^2 \right]$ is a constant independent of $\phi$.

**Diffusion with $\epsilon$-Prediction.** Similarly, we have the $\epsilon$-prediction loss (Ho et al., 2020a):

$$\mathbb{E}_{x_0 \sim p(x_0 \mid x_t)} \left[ \|\epsilon_\phi(x_t, t) - \epsilon\|_2^2 \right] = \frac{\alpha_t^2}{\sigma_t^2} L_\phi(x_t), \tag{7}$$

whose optimal solution is the conditional expectation of the noise added into $x_t$:

$$\epsilon_{\phi^*}(x_t, t) = \mathbb{E}[\epsilon \mid x_t] = \frac{x_t - \alpha_t f_{\phi^*}(x_t, t)}{\sigma_t}.$$

**Diffusion with $v$-Prediction.** For the $v$-prediction loss (Salimans & Ho, 2022):

$$\mathbb{E}_{x_0 \sim p(x_0 \mid x_t)} \left[ \|v_\phi(x_t, t) - (\alpha_t \epsilon - \sigma_t x_0)\|_2^2 \right] = \frac{(\alpha_t^2 + \sigma_t^2)^2}{\sigma_t^2} L_\phi(x_t), \tag{8}$$

the optimal solution is

$$v_{\phi^*}(x_t, t) = \mathbb{E}[\alpha_t \epsilon - \sigma_t x_0 \mid x_t] = \alpha_t \epsilon_{\phi^*}(x_t, t) - \sigma_t f_{\phi^*}(x_t, t) = \frac{\alpha_t x_t - (\alpha_t^2 + \sigma_t^2) f_{\phi^*}(x_t, t)}{\sigma_t}.$$

**Rectified Flow.** In rectified flow (Liu et al., 2022b; Lipman et al., 2022), the objective expressed as

$$\mathbb{E}_{x_0 \sim p(x_0 \mid x_t)} \left[ \|v_\phi^{\text{FM}}(x_t, t) - (\epsilon - x_0)\|_2^2 \right] = \sigma_t^{-2} L_\phi(x_t) \tag{9}$$

is referred to as a velocity-prediction loss, whose optimal solution is

$$v_{\phi^*}^{\text{FM}}(x_t, t) = \mathbb{E}[\epsilon - x_0 \mid x_t] = \epsilon_{\phi^*}(x_t, t) - f_{\phi^*}(x_t, t) = \frac{x_t - (\alpha_t + \sigma_t) f_{\phi^*}(x_t, t)}{\sigma_t}$$

$$= \frac{(\sigma_t - \alpha_t) x_t + (\alpha_t + \sigma_t) v_{\phi^*}(x_t, t)}{\alpha_t^2 + \sigma_t^2}. \tag{10}$$

For rectified flow, it is conventional to set $\sigma_t = t$ and $\alpha_t = 1 - t$, under which the identities hold:

$$v_{\phi^*}^{\text{FM}}(x_t, t) = \frac{x_t - f_{\phi^*}(x_t, t)}{t} = \frac{\epsilon_{\phi^*}(x_t, t) - x_t}{1 - t} = \frac{(2t-1)x_t + v_{\phi^*}(x_t, t)}{t^2 + (1-t)^2} = -\frac{x_t + t S_{\phi^*}(x_t, t)}{1 - t}. \tag{11}$$

This also implies that, in rectified flow, $f_{\phi^*}(x_t, t) = x_t - t v_{\phi^*}^{\text{FM}}(x_t, t)$.

**TrigFlow.** In TrigFlow (Lu & Song, 2024), the data corruption process is modified to

$$x_{t_{\text{Trig}}} = \cos(t_{\text{Trig}})\sigma_d x_0 + \sin(t_{\text{Trig}})\sigma_d \epsilon,$$

and the corresponding loss becomes

$$L_{\phi,\mathrm{Trig}}(x_{t_{\mathrm{Trig}}}) = \mathbb{E}_{p(x_0 \mid x_{t_{\mathrm{Trig}}})} \left[ \left\| \sigma_d F_\phi(x_{t_{\mathrm{Trig}}}, t_{\mathrm{Trig}}) - (\cos(t_{\mathrm{Trig}})\sigma_d\epsilon - \sin(t_{\mathrm{Trig}})\sigma_d x_0) \right\|_2^2 \right].$$

As in SANA-Sprint (Chen et al., 2025), to make $\frac{(1-t)^2}{t^2} = \frac{\cos^2(t_{\mathrm{Trig}})}{\sin^2(t_{\mathrm{Trig}})}$, we set $t = \frac{\sin(t_{\mathrm{Trig}})}{\sin(t_{\mathrm{Trig}})+\cos(t_{\mathrm{Trig}})}$, $\sin(t_{\mathrm{Trig}}) = \frac{t}{\sqrt{t^2+(1-t)^2}}$, $\cos(t_{\mathrm{Trig}}) = \frac{1-t}{\sqrt{t^2+(1-t)^2}}$, resulting in a $v$-prediction loss with $\alpha_{t_{\mathrm{Trig}}} = \frac{1-t}{\sqrt{t^2+(1-t)^2}}$ and $\sigma_{t_{\mathrm{Trig}}} = \frac{t}{\sqrt{t^2+(1-t)^2}}$. Denoting $x_t = \frac{\sqrt{t^2+(1-t)^2}}{\sigma_d} x_{t_{\mathrm{Trig}}}$, we have

$$\frac{L_{\phi,\mathrm{Trig}}(x_{t_{\mathrm{Trig}}})}{\sigma_d^2} = \frac{t^2+(1-t)^2}{t^2} L_\phi(x_t) = (t^2 + (1-t)^2)\mathbb{E}_{x_0 \sim p(x_0 \mid x_{t_{\mathrm{Trig}}})} \left[ \left\| v_\phi^{\mathrm{FM}}(x_t, t) - (\epsilon - x_0) \right\|_2^2 \right].$$

### 2.3 A Unified Perspective via Loss Reweighting

The relationships among these quantities, which are linear transformations of one another given $x_t$, can be summarized by expressing the optimal score function $S_{\phi^*}(x_t, t)$ in multiple equivalent forms:

$$S_{\phi^*}(x_t, t) = \begin{cases} -\frac{x_t - \alpha_t f_{\phi^*}(x_t, t)}{\sigma_t^2} & (x_0\text{-prediction}) \\ -\frac{\epsilon_{\phi^*}(x_t, t)}{\sigma_t} & (\epsilon\text{-prediction}) \\ -\frac{\sigma_t x_t + \alpha_t v_{\phi^*}(x_t, t)}{\sigma_t(\alpha_t^2 + \sigma_t^2)} & (v\text{-prediction}) \\ -\frac{x_t + \alpha_t v_{\phi^*}^{\mathrm{FM}}(x_t, t)}{\sigma_t(\alpha_t + \sigma_t)} = -\frac{x_t + (1-t)v_{\phi^*}^{\mathrm{FM}}(x_t, t)}{t} & (\text{flow matching}) \end{cases} \tag{12}$$

It is now clear that whether one uses $x_0$-, $\epsilon$-, or $v$-prediction in diffusion, or velocity-prediction in rectified flow or TrigFlow, all approaches optimize the same underlying objective, differing only in how each timestep $t \sim p(t)$ is weighted in the overall loss. Although these weightings do not affect the optimal solution for any fixed $t$ in theory, in practice both the timestep distribution $p(t)$ and any additional factor $w_t$ determine which timesteps exert greater influence on optimizing the shared parameter set $\phi$. More specifically, letting $L_{\phi,t} = \mathbb{E}_{x_t \sim p(x_t)}[L_\phi(x_t)]$, the overall loss for pretraining a diffusion or flow-matching model can be written as

$$L_\phi = \mathbb{E}_{t \sim p(t)}\mathbb{E}_{x_t \sim p(x_t)} \left[ w_t \cdot \frac{\alpha_t^2}{\sigma_t^2} L_\phi(x_t) \right] = \int w_t p(t) \cdot \frac{\alpha_t^2}{\sigma_t^2} L_{\phi,t} \, dt = C_\pi \cdot \mathbb{E}_{t \sim \pi(t)} \left[ \frac{\alpha_t^2}{\sigma_t^2} L_{\phi,t} \right], \tag{13}$$

where $C_\pi = \int w_t \, p(t) \, dt = \mathbb{E}_{p(t)}[w_t]$ is a constant independent of $\phi$, and

$$\pi(t) = \frac{w_t p(t)}{\int w_t \, p(t) \, dt} \tag{14}$$

is the weight-normalized distribution of $t$. For example, in DDPM (Ho et al., 2020a) we have $w_t = 1$, so $\pi(t) = p(t)$; in rectified flow we have $w_t = (1-t)^{-2}$, giving $\pi(t) = \frac{(1-t)^{-2}p(t)}{\int (1-t)^{-2}p(t)\,dt}$.

Thus, any claim that a particular $w_t$ is superior without controlling for $p(t)$ may be misleading, since the expected loss depends jointly on both. To illustrate, Figure 2 shows, for each column, the resulting distribution $\pi(t)$ when a typical $p(t)$—determined by the noise schedule—is combined with different $w_t$. Notably, even when $w_t$ and $p(t)$ differ substantially, the resulting $\pi(t)$ distributions can look quite similar. A more detailed description of Figure 2 is provided in Appendix C.

In summary, Gaussian-based diffusion and flow matching models share the same theoretical optimal solutions. Their practical differences arise from the weight-normalized timestep distribution, as shown in (14). This insight supports the extension of diffusion distillation techniques—originally developed for diffusion models—to flow matching models, with the caveat that one must account for the differences in their respective weight-normalized timestep distributions, $\pi(t)$.

## 3 Score Distillation of DiT-based Flow-Matching Models

Diffusion distillation typically relies on access to the teacher's score estimates or $x_0$-predictions given $x_t$, which are readily available from pretrained diffusion models. These quantities can also

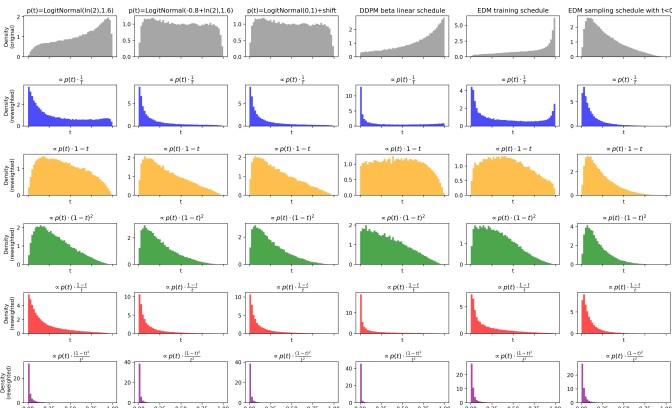

Figure 2: The first row shows density plots of various noise schedules mapped to $t \in (0,1)$ by aligning their signal-to-noise ratio (SNR), $\text{SNR}_t = \alpha_t^2/\sigma_t^2$, with $(1-t)^2/t^2$, which corresponds to setting $t = 1/(1+\sqrt{\text{SNR}_t})$. The remaining rows show the weight-normalized distribution of $t$ under different weighting schemes: $1/t$, $1-t$, $(1-t)^2$, $(1-t)/t$, and $(1-t)^2/t^2$. The first column corresponds to the default schedule used in this paper and in TrigFlow training of SANA-Sprint; the second to the default TrigFlow schedule; the third to the discretized schedule of SANA; the fourth to the DDPM beta linear schedule; the fifth to EDM's training schedule; and the sixth to EDM's sampling schedule restricted to $t < 0.8$, as in SiD for score distillation.

be obtained from velocity predictions in flow-matching models via a simple linear relation between the predicted velocity and $x_t$. Specifically, for T2I flow-matching models, as shonw in (11), if $v_\phi^{\text{FM}}(x_t, t, c)$ denotes the estimated velocity given $x_t$ and text condition $c$, then the teacher's $x_0$-prediction $\mathbb{E}[x_0 \mid x_t, c]$ can be approximated as

$$f_\phi(x_t, t, c) = x_t - t v_\phi^{\text{FM}}(x_t, t, c).$$

Classifier-free guidance (CFG, Ho & Salimans (2022)) is critical for strong T2I performance. Unless otherwise noted, we redefine $f_\phi(x_t, t, c)$ under CFG with a scale of 4.5:

$$f_\phi(x_t, t, c) = \left(x_t - t v_\phi^{\text{FM}}(x_t, t, \emptyset)\right) + 4.5\left[\left(x_t - t v_\phi^{\text{FM}}(x_t, t, c)\right) - \left(x_t - t v_\phi^{\text{FM}}(x_t, t, \emptyset)\right)\right]. \quad (15)$$

To distill the pretrained teacher, we adopt Fisher divergence minimization, extending the few-step SiD method (Zhou et al., 2025a) into **SiD-DiT**. A four-step generator is defined as

$$x_g^{(k)} = G_\theta\left((1-t_k)\,\text{sg}(x_g^{(k-1)}) + t_k z_k,\, t_k,\, c\right), \quad t_k = \left(1 - \tfrac{k-1}{4}\right)T, \quad z_k \sim \mathcal{N}(0, \mathbf{I}), \quad (16)$$

where $\text{sg}(\cdot)$ is the stop-gradient operator, $T = 1000$, and $k = 1, 2, 3, 4$.

We sample $k \in \{1, 2, 3, 4\}$ uniformly and $t \sim p(t)$, and forward-diffuse $x_g^{(k)}$ as

$$x_t^{(k)} = (1-t_k)\,x_g^{(k)} + t_k \epsilon_k, \quad \epsilon_k \sim \mathcal{N}(0, \mathbf{I}). \quad (17)$$

Operating in a data-free manner, SiD-DiT alternates between updating $\theta$ given $\psi$ (a "fake" flow-matching network) and updating $\psi$ given $\theta$. The fake network $f_\psi$ is initialized from $f_\phi$ and trained on a uniform mixture of $x_g^{(k)}$ across the four generation steps using a flow-matching loss. The generator loss is defined as

$$L_\theta(x_t^{(k)}) = w_t\left(f_\phi(x_t^{(k)}, t_k, c) - f_\psi(x_t^{(k)}, t_k, c)\right)^T\left(f_\psi(x_t^{(k)}, t_k, c) - x_g^{(k)}\right), \quad (18)$$

where $w_t$ is a weighting factor, set to $1-t$ by default. We apply CFG with a scale of 4.5 to $f_\psi$ during both its own training and the update of $\theta$, following the long-and-short guidance (LSG) strategy of Zhou et al. (2025b).

When additional data are available, we incorporate the Diffusion GAN (Wang et al., 2023a) adversarial loss, steering generation toward the target distribution. Unlike adversarial enhancement in SiD for U-Net, where the encoder–decoder architecture provides a natural bottleneck for extracting discriminator features via channel pooling (Zhou et al., 2025c), DiT backbones lack such a bottleneck. We empirically find that pooling along the spatial dimension after the final normalization layer but before the projection and unpatchifying layers provides an effective discriminator feature representation. This strategy is simple, effective, and introduces no additional parameters.

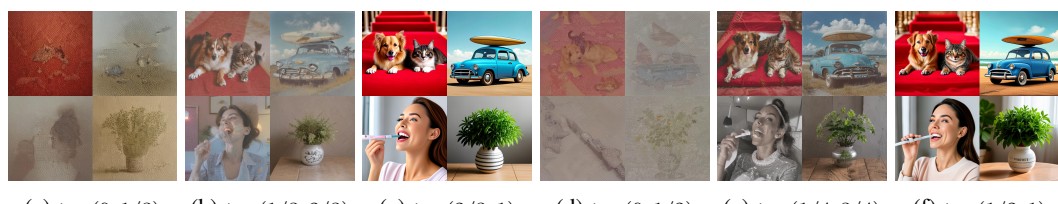

(a) $t \in (0, 1/3)$    (b) $t \in (1/3, 2/3)$    (c) $t \in (2/3, 1)$    (d) $t \in (0, 1/2)$    (e) $t \in (1/4, 3/4)$    (f) $t \in (1/2, 1)$

Figure 3: Comparison of distilled `Sana_600M_512px_diffusers` by restricting $t$ to different ranges. The text prompts are: 'a dog and a cat laying on the red carpet on the floor.', 'an old blue car with a surfboard on top', 'a lady is about to put an automatic tooth brush in her mouth', and 'a good luck plant is in a round vase.'

## 4 EXPERIMENTAL RESULTS

We conduct comprehensive experiments across DiT-based flow-matching models with varying architectures, noise schedules, and model sizes, showcasing the efficiency and robustness of SiD-DiT. All experiments, except for FLUX1.DEV at $1024 \times 1204$ resolution, are conducted on a single node equipped with eight A100 or H100 GPUs (each with 80GB memory). Initial development employs AMP (via `torch.autocast`) together with Fully Sharded Data Parallel (FSDP), which provides robust performance on SANA-0.6B/1.6B, SD3-MEDIUM, and SD3.5-MEDIUM. However, this configuration runs into memory limitations for larger models such as SD3.5-LARGE and FLUX1.DEV, where CPU offloading becomes necessary but significantly slows training.

To overcome this bottleneck, we switch to a pure BF16-based distillation pipeline. BF16 achieves higher throughput and lower memory usage but requires more aggressive settings—specifically, a learning rate of $10^{-5}$ and Adam $\epsilon = 10^{-4}$—to avoid gradient underflow. While other parameterizations are possible, this setting suffices for all DiT models in this paper. In addition, we decouple the main training loop from the VAE and text encoder, which are periodically loaded to preprocess text prompts—and optionally real images—in a streaming fashion. These enhancements enable effective distillation of both SD3.5-LARGE and FLUX1.DEV under the same hardware constraints.

We begin with the lightweight SANA (Xie et al., 2025) as a case study, leveraging publicly available checkpoints trained under both Rectified Flow and TRIGFLOW. In contrast to SANA-Sprint, which requires real data and can only distill TrigFlow-based checkpoints, SiD-DiT operates entirely without real data, enabling fully data-free distillation for both formulations. This provides a more faithful assessment of teacher–student knowledge transfer, free from the confounding effects of downstream fine-tuning, and establishes a broadly applicable distillation framework.

We further extend SiD-DiT with adversarial learning. For this variant, we incorporate additional data from MidJourney-v6-llava , a fully synthesized dataset that ensures reproducibility without copyright or licensing concerns. We denote this variant as $SiD_2^a$, which initializes from a SiD-distilled generator and continues training with an additional DiffusionGAN-based adversarial loss.

While the quality of this dataset is limited, it demonstrates that the utilization of additional data can increase sample diversity, improving FID. However, it does not substantially enhance visual quality, and the MJ-style generations it induces may not align with user preferences. We therefore recommend its use only for evaluation purposes, while emphasizing that high-quality real data is preferable when adversarial learning is employed.

Finally, we evaluate both the data-free and adversarial variants on additional flow-matching models, adapting the codebase to their architectural specifics. Notably, only minimal hyperparameter tuning and model-specific customization are required, as summarized in Tables 4 and 5. As shown in Figure 4, SiD-DiT achieves rapid improvements in both FID and CLIP scores during distillation across all nine DiT models. Full implementation details are provided in the supplementary code release.

### 4.1 UNDERSTANDING THE ROLE OF LOSS REWEIGHTING

We first examine three extreme forms of loss reweighting, where the generator loss is restricted to one of three disjoint intervals: $t \in (0, \frac{1}{3})$, $t \in (\frac{1}{3}, \frac{2}{3})$, or $t \in (\frac{2}{3}, 1)$. Qualitative generations are shown in Figure 3(a–c).

We find that restricting to $t \in (\frac{2}{3}, 1)$ is sufficient to produce visually appealing images, though these often lack high-frequency details and diversity. In contrast, $t \in (\frac{1}{3}, \frac{2}{3})$ yields finer detail but with a duller, hazier appearance, while restricting to $t \in (0, \frac{1}{3})$ fails to produce reasonable generations. We then consider less extreme reweighting with partially overlapping intervals: $t \in (0, \frac{1}{2})$, $t \in (\frac{1}{4}, \frac{3}{4})$, and $t \in (\frac{1}{2}, 1)$. The corresponding qualitative results are shown in Figure 3(d–f). Similar trends are observed, though the effects are less pronounced.

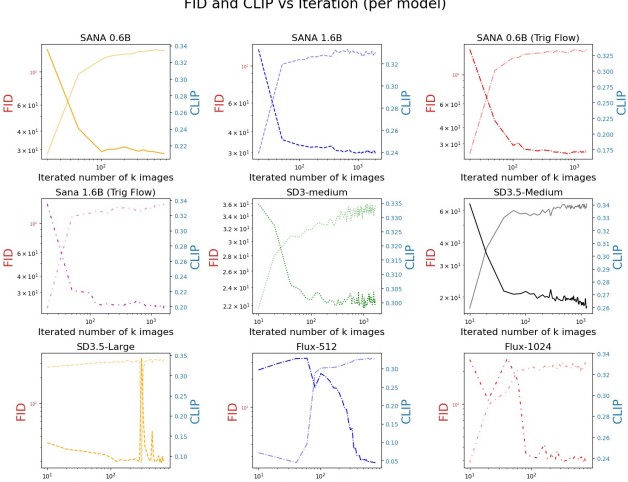

These empirical findings provide intuition for designing $p(t)$ and $w_t$. Since the effective timestep distribution $\pi(t)$ depends only on their product, adjusting both is not strictly necessary to preserve the loss structure. In this paper, we fix $p(t) = \text{Logit}\,\mathcal{N}(t; \ln 2, 1.6^2)$ to match the schedule used for finetuning the SANA-Sprint teacher. We set $w_t = 1 - t$. The resulting weight-normalized distribution $\pi(t)$ is shown in the first column, third row of Fig-

Figure 4: This plot shows the evolution of FID (solid lines, left y-axis) and CLIP score (matching line styles with reduced opacity, right y-axis) as a function of the number of iterated images (in thousands) for SiD-DiT. Because the x-axis is log-scaled, the near-linear trends in many panels reflect a rapid initial decline in FID accompanied by a corresponding rise in CLIP score, followed by progressively smaller gains as training continues. This consistent behavior across architectures and model sizes shows that SiD-DiT quickly improves both image fidelity and semantic alignment during the early stages of distillation.

ure 2. While a systematic study of how varying $p(t)$ and $w(t)$ affects performance is beyond the scope of this paper, our observation is consistent with Figure 2: stronger emphasis on larger $t$ values (heavier noise) produces visually appealing but less detailed images, and smaller $t$ highlights fine-grained detail at the cost of vividness. Overall, the chosen combination of $p(t)$ and $w_t$ yields a $\pi(t)$ with full coverage over $t$, which we find to perform well across all T2I flow-matching models tested in this paper.

## 4.2 DISTILLATION OF FLOW-MATCHING-BASED SANA MODELS

We apply SiD-DiT to SANA and compare it against both SANA and SANA-SPRINT. Unlike SANA-SPRINT, which requires finetuning rectified flow checkpoints into TrigFlow, SiD-DiT is natively compatible with both frameworks. In practice, the same SiD-DiT code used for TrigFlow can be applied to rectified flow SANA by simply scaling the time variable $t$ by 1000.

**Rectified-Flow SANA:** We evaluate two rectified-flow checkpoints: Sana_600M_512px_diffusers and Sana_1600M_512px_diffusers. These models cannot be distilled by SANA-SPRINT without prior adaptation, whereas SiD-DiT can be applied directly. **TrigFlow SANA:** We also evaluate two TrigFlow checkpoints: SANA_Sprint_0.6B_1024px_teacher_diffusers and SANA_Sprint_1.6B_1024px_teacher_diffusers. Both are finetuned under TrigFlow to enable SANA-SPRINT, whereas SiD-DiT applies directly, either with or without teacher finetuning.

Quantitative results for both rectified-flow- and TrigFlow-based SANA are reported in Table 1. We evaluate performance on the SANA backbone using zero-shot FID, CLIP score (Radford et al., 2021), and GenEval (Ghosh et al., 2023), with FID and CLIP computed on the 10k COCO-2014 validation subset employed by DMD2 (Yin et al., 2024a). We also evaluate human preference using LAION Aesthetics (Schuhmann et al., 2021), HPSv2 (Wu et al., 2023), ImageReward (Xu et al., 2023), and PickScore (Kirstain et al., 2023) on 2048 Pick-a-Pic (Kirstain et al., 2023) validation prompts.

Table 1: Comparison of SiD-DiT, SiD$_2^a$-DiT, and SANA/SANA-Sprint in performance and efficiency. **Bold** indicates the best score.

| Model | #Steps | Params (B) | FID ↓ | CLIP ↑ | GenEval ↑ | Aesth. ↑ | HPSv2 ↑ | ImgRwd ↑ | PickScore ↑ |
|---|---|---|---|---|---|---|---|---|---|
| **SANA 0.6B** | | | | | | | | | |
| SANA (Xie et al., 2024) | 20 | 0.6 | 28.01 | 0.329 | 0.641 | **6.320** | 0.287 | 1.111 | **22.125** |
| SiD-DiT (SANA) | 4 | 0.6 | 29.43 | **0.333** | **0.652** | 6.168 | **0.306** | **1.117** | 21.685 |
| SiD$_\alpha^2$-DiT (SANA) | 4 | 0.6 | **25.82** | 0.330 | 0.643 | 6.160 | 0.305 | 1.111 | 21.666 |
| **SANA 1.6B** | | | | | | | | | |
| SANA (Xie et al., 2024) | 20 | 1.6 | 28.71 | 0.328 | 0.655 | 6.151 | 0.306 | 1.254 | 21.984 |
| SiD-DiT (SANA) | 4 | 1.6 | 26.94 | **0.331** | **0.670** | **6.245** | **0.317** | **1.283** | 21.883 |
| SiD$_\alpha^2$-DiT (SANA) | 4 | 1.6 | **26.31** | **0.331** | 0.665 | 6.185 | 0.308 | 1.092 | **22.035** |
| **SANA TrigFlow 0.6B** | | | | | | | | | |
| SANA Sprint Teacher | 20 | 0.6 | 25.64 | 0.335 | **0.780** | 6.222 | 0.299 | **1.115** | 21.78 |
| SANA Sprint (Chen et al., 2025) (TrigFlow, 1 step) | 1 | 0.6 | 24.60 | 0.336 | 0.770 | **6.361** | 0.286 | 1.006 | 21.805 |
| SANA Sprint (Chen et al., 2025) (TrigFlow, 4 steps) | 4 | 0.6 | 26.32 | 0.335 | 0.766 | 6.325 | 0.301 | 1.111 | 22.125 |
| SiD-DiT (SANA, TrigFlow) | 4 | 0.6 | 25.81 | **0.340** | 0.763 | 6.243 | **0.308** | 1.049 | 21.561 |
| SiD$_\alpha^2$-DiT (SANA, TrigFlow) | 4 | 0.6 | **22.46** | 0.330 | 0.772 | 6.188 | 0.295 | 0.924 | 21.625 |
| **SANA TrigFlow 1.6B** | | | | | | | | | |
| SANA Sprint Teacher | 20 | 1.6 | 25.64 | 0.335 | **0.776** | 6.209 | 0.304 | **1.163** | 21.93 |
| SANA Sprint (Chen et al., 2025) (TrigFlow, 1 step) | 1 | 1.6 | 24.60 | 0.335 | 0.768 | **6.362** | 0.293 | 1.030 | 22.006 |
| SANA Sprint (Chen et al., 2025) (TrigFlow, 4 steps) | 4 | 1.6 | 24.79 | 0.335 | 0.768 | 6.338 | 0.300 | 1.081 | 22.123 |
| SiD-DiT (SANA, TrigFlow) | 4 | 1.6 | 23.81 | **0.340** | 0.774 | 6.305 | **0.307** | 1.102 | 21.897 |
| SiD$_\alpha^2$-DiT (SANA, TrigFlow) | 4 | 1.6 | **22.58** | 0.335 | 0.768 | 6.200 | 0.303 | 1.073 | 21.936 |

Table 2: Comparison of SiD-DiT, SiD$_2^a$-DiT, and SD3/FLUX baselines in performance and efficiency. **Bold** indicates the best score within each block.

| Model | #Steps | Params (B) | FID ↓ | CLIP ↑ | Aesth. ↑ | HPSv2 ↑ | ImgRwd ↑ | PickScore ↑ |
|---|---|---|---|---|---|---|---|---|
| **SD3-Medium** | | | | | | | | |
| SD3-Medium (base) | 28 | 2.0 | 24.40 | 0.336 | 5.870 | 0.297 | **1.051** | 21.574 |
| Flash SD3 (Chadebec et al., 2025) | 4 | 2.0 | 22.70 | 0.338 | 5.820 | 0.289 | 0.997 | 21.326 |
| SiD-DiT (SD3-Medium) | 4 | 2.0 | 22.05 | **0.341** | **6.054** | 0.301 | 1.017 | 21.686 |
| SiD$_\alpha^2$-DiT (SD3-Medium) | 4 | 2.0 | **21.64** | 0.327 | 6.050 | **0.305** | 1.022 | **21.836** |
| **SD3.5-Medium** | | | | | | | | |
| SD3.5-Medium (base) | 40 | 2.5 | 22.51 | **0.342** | 5.973 | 0.300 | 1.089 | 21.974 |
| SD3.5-Medium-Turbo | 8 | 2.5 | 21.15 | 0.337 | 5.971 | 0.263 | 0.633 | 21.478 |
| SiD-DiT (SD3.5-Medium) | 4 | 2.5 | 21.07 | 0.340 | **6.187** | **0.308** | **1.097** | **22.037** |
| SiD$_\alpha^2$-DiT (SD3.5-Medium) | 4 | 2.5 | **20.92** | 0.331 | 6.077 | 0.291 | 0.967 | 21.870 |
| **SD3.5-Large** | | | | | | | | |
| SD3.5-Large (base) | 28 | 8.1 | **20.81** | **0.341** | 6.097 | 0.305 | 1.127 | 22.245 |
| SD3.5-Turbo-Large | 4 | 8.1 | 26.11 | 0.340 | **6.198** | 0.302 | 1.086 | 22.171 |
| SiD-DiT (SD3.5-Large) | 4 | 8.1 | 21.10 | **0.341** | 6.132 | 0.309 | 1.214 | 22.097 |
| SiD$_\alpha^2$-DiT (SD3.5-Large) | 4 | 8.1 | 22.10 | 0.337 | 6.167 | **0.316** | **1.275** | **22.407** |
| **FLUX-1 Family** | | | | | | | | |
| FLUX-1-Dev (base) | 28 | 12.0 | **22.89** | 0.344 | 6.184 | 0.297 | 0.897 | 21.862 |
| Flux-Schnell (Black Forest Labs) | 4 | 12.0 | 23.42 | **0.345** | 6.173 | 0.302 | 1.109 | 21.796 |
| FLUX-1-Turbo (Black Forest Labs) | 4 | 12.0 | 24.92 | 0.332 | 6.192 | 0.302 | 1.012 | 21.977 |
| Hyper-FLUX (ours) | 4 | 12.0 | 25.44 | 0.332 | **6.257** | **0.310** | 1.028 | **22.090** |
| SiD-DiT (FLUX-1-Dev) | 4 | 12.0 | 27.86 | 0.330 | 5.964 | 0.305 | **1.203** | 21.583 |

For rectified-flow SANA (0.6B and 1.6B), SiD-DiT achieves comparable FID to the original teacher while slightly improving CLIP and maintaining GenEval. With adversarial learning, SiD$_2^a$ reduces FID substantially (25.82 vs. 28.01 at 0.6B, and 26.31 vs. 28.71 at 1.6B) while preserving CLIP and GenEval scores. For TrigFlow-based SANA, SiD outperforms SANA-Sprint across both scales. At 0.6B, SiD improves FID from 26.97 to 25.34, and further down to 22.46 with adversarial learning, while maintaining higher CLIP and GenEval scores. At 1.6B, SiD reduces FID from 24.60 to 23.81 (and 22.58 with SiD$_2^a$) and also achieves the best CLIP (0.336) without sacrificing GenEval (0.77). SiD also achieves competitive human preference performance relative to both the teacher model and other distillation baselines for both rectified-flow and TrigFlow-based SANA. On rectified-flow SANA, SiD notably surpasses the teacher in HPSv2 (0.287 vs. 0.306 at 0.6B, and 0.306 vs. 0.317 at 1.6B), while maintaining comparable performance on the other preference metrics.

Overall, SiD-DiT delivers consistent improvements over SANA-Sprint on TrigFlow checkpoints, despite being data-free, while SiD$_2^a$-DiT provides the strongest FID reductions across all settings. These results underscore the robustness of our method in both data-free and data-aided distillation.

### 4.3 DISTILLATION OF MMDIT MODELS (SD3-MEDIUM, SD3.5-MEDIUM, SD3.5-LARGE)

We evaluate SiD-DiT on SD3-MEDIUM (2B parameters) and SD3.5-MEDIUM (2.5B parameters), both based on the MMDiT architecture (Esser et al., 2024), which improves visual fidelity, typogra-

phy, complex prompt comprehension, and computational efficiency. Using the same teacher noise schedule as SANA and the $w_t = 1 - t$ reweighting, we observe consistent success across both models, with results summarized in Table 2.

On SD3-MEDIUM, SiD-DiT matches the teacher in FID and CLIP, while the adversarial variant SiD$_2^a$-DiT achieves a substantial FID reduction to **21.64**. On SD3.5-MEDIUM, SiD-DiT not only surpasses the teacher but also outperforms SD-Turbo, with SiD$_2^a$-DiT delivering the best FID of 20.92, LAION Aesthetics of 6.187, HPSv2 of 0.308 (Sauer et al., 2024; Chadebec et al., 2025). These results underscore the robustness of SiD-DiT as a data-free framework, while demonstrating that adversarial training with additional data can further enhance performance via Diffusion GAN.

Building on these successes, we extend SiD-DiT to SD3.5-LARGE (8.1B parameters), the largest open-source MMDiT model currently available in the Stable Diffusion family and more than three times larger than SD3.5-MEDIUM (2.5B). Scaling to this size introduces substantial memory challenges; however, our FSDP+FP16+streaming strategy alleviates these constraints, enabling distillation on a single 8×80GB A100/H100 node without CPU offloading. As shown in Table 2, SiD-DiT achieves an FID of **20.57**, substantially outperforming SD3.5-Turbo-Large (26.11) and slightly surpassing the teacher baseline (20.81). Its CLIP score (0.341) matches that of the teacher. For human preference, SiD-DiT surpasses the teacher on LAION Aesthetics, HPSv2 and Image Reward, and can even outperform teacher in all 4 perference metrics with SiD$_\alpha^2$-DiT. These results demonstrate that SiD-DiT scales effectively to large MMDiTs, providing a practical, out-of-the-box solution for distilling models at this scale.

### 4.4 DISTILLATION OF FLUX.1-DEV

The SiD-DiT framework delivers competitive generation quality and serves as an out-of-the-box DiT distillation method that is robust across diverse architectures. In our implementation, SiD-DiT employs CFG as formalized in Equation (15), consistent with the Stable Diffusion T2I family. In contrast, FLUX.1-DEV adopts a learned guidance embedding by default and does not provide an explicit unconditional branch for CFG. We partially attribute the modest performance gap of SiD-DiT on FLUX.1-DEV to this guidance-mechanism mismatch. Importantly, we did not introduce any Flux-specific modifications beyond the minimal adjustments required to make the model runnable. Even under this direct application, SiD-DiT achieves strong qualitative results (Figure 5) and competitive quantitative metrics (Table 2), while efficiently distilling the 12B-parameter FLUX.1-DEV model at $512 \times 512$ resolution on a single node with eight 80GB GPUs, and at $1024 \times 1024$ resolution on a single node with eight 192GB GPUs. Further improvements are likely possible by tailoring SiD-DiT more closely to the unique design of FLUX.1-DEV, for example by integrating its learned guidance embeddings into the distillation objective or developing a hybrid approach that blends CFG with model-specific guidance. Such targeted extensions may help close the remaining performance gap and demonstrate the flexibility of SiD-DiT across emerging flow-matching architectures.

## 5 CONCLUSION

In this work, we revisited the theoretical foundations of diffusion and flow matching models, showing that under Gaussian assumptions, their optimal solutions are equivalent despite differences in loss weighting and practical implementations. Building on this unified perspective, we demonstrated that score distillation—originally developed for diffusion models—can be effectively and robustly extended to flow matching models without requiring model-specific adaptations or teacher finetuning. Through the use of few-step Score identity Distillation (SiD), we successfully distilled a wide range of pretrained text-to-image flow matching models, including SANA, SD3, SD3.5, and FLUX.1-dev, into efficient four-step generators. Our approach uses a single, shared codebase and training configuration across models of varying architectures and parameter scales, showcasing the generality and stability of score distillation in this new context. These findings not only clarify misconceptions in prior work regarding the applicability of score distillation to flow-based models, but also open new directions for compressing and accelerating modern text-to-image generators. By bridging the gap between diffusion and flow matching, our work provides a solid theoretical and empirical foundation for future research on unified generative modeling and fast sampling strategies.

## REPRODUCIBILITY STATEMENT

We release the full anonymized codebase and training scripts in the supplementary material to facilitate reproduction of all experiments. All algorithmic derivations are detailed in the main text, while hyperparameter settings and precision configurations (AMP vs. BF16) are reported in Tables 4 and 5 of the Appendix.

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

## A  USE OF LARGE LANGUAGE MODELS (LLMS)

Large Language Models (LLMs) were used to improve grammar, clarity, and readability of the text. They also assisted with code debugging, annotation, and anonymization.

## B  RELATED WORK

Acceleration strategies for pretrained diffusion models generally fall into two categories: training-free methods and diffusion distillation. Training-free methods, such as DDIM (Song et al., 2020), DPM-Solver (Lu et al., 2022), and EDM Heun's sampler (Karras et al., 2022), reduce the number of function evaluations (NFEs) without retraining. These approaches have successfully lowered NFEs from hundreds to just a few dozen, although performance typically degrades when NFEs drop below 20.

Diffusion distillation, on the other hand, leverages the estimated score function from pretrained models to train faster generators (Luhman & Luhman, 2021; Salimans & Ho, 2022; Meng et al., 2023). It comprises two main branches: trajectory distillation (Song et al., 2023; Song & Dhariwal, 2023; Luo et al., 2023a; Kim et al., 2023), which requires access to real or teacher-synthesized data, and score distillation (Poole et al., 2023; Wang et al., 2023b; Luo et al., 2023c; Yin et al., 2024b; Nguyen & Tran, 2024; Zhou et al., 2024), which can be performed in a data-free setting but may also benefit from using real or synthetic data. Some score distillation methods, such as Diff-Instruct (Luo et al., 2023c) and SiD (Zhou et al., 2024; 2025b), are designed to operate without real data, while others require access to real or teacher-synthesized data (Yin et al., 2024b;a; Sauer et al., 2023), or are enhanced by incorporating such data (Zhou et al., 2025c).

A wide variety of score distillation methods can be used to distill the teacher model into one or few-step T2I generators, such as DMD (Yin et al., 2024b;a) and SwiftBrush (Nguyen & Tran, 2024) that are based on minimizing the KL divergence between the generator's distribution in the diffused space and the data distribution in the diffused space estimated by the teacher (Poole et al., 2023; Wang et al., 2023b; Luo et al., 2023c). One can also utilize other divergence, including Fisher divergence (Zhou et al., 2024; 2025c;b;a), a variant of Fisher divgernce (Luo et al., 2024), and f-divergence (Xu et al., 2025).

Flow matching has recently emerged as a promising alternative for generative modeling (Liu et al., 2022b; Lipman et al., 2022; Albergo et al., 2023). A key example is *rectified flow* (Liu et al., 2022b), also known as flow matching with an optimal transport path (Lipman et al., 2022). Rectified flow encourages straighter trajectories between noise and data, reducing the number of function evaluations (NFEs) needed for sampling and enabling one- or few-step generation via ReFlow (Liu et al., 2022b). Another representative approach is *TrigFlow* (Lu & Song, 2024), now the preferred framework for continuous consistency distillation and successfully applied by Chen et al. (2025) to develop SANA-Sprint, which distills SANA T2I models after finetuning rectified flow teachers into TrigFlow. In contrast, our method works directly with SANA models trained under either rectified flow or TrigFlow, without requiring such finetuning.

Although originally proposed as a faster and simpler alternative to diffusion, recent theoretical insights have shown that, under Gaussian assumptions, rectified flow is fundamentally equivalent to diffusion: training a Gaussian noise-based rectified flow model is mathematically equivalent to training a Gaussian diffusion model, and their corresponding SDE/ODE sampling procedures are interchangeable (Albergo et al., 2023; Kingma & Gao, 2023; Ma et al., 2024; Gao et al., 2024; Geffner et al., 2025), and thus the distillation techniques proposed for diffusion models can be adapted to Gaussian-based rectified flow, such as consistency models (Yang et al., 2024; Lu & Song, 2024). Nevertheless, practical differences remain, such as in noise schedules, loss formulations, and network architectures.

Although score distillation has proven highly effective in reducing diffusion models to one- or few-step generators (Luo et al., 2023c; Zhou et al., 2024; Yin et al., 2024c;a; Zhou et al., 2025c), its application to flow matching remains largely unexplored. Methods like ReFlow construct noise-image pairs by solving a pretrained flow model's ODE and then use these pairs to train a fast generator. Rectified flow is often considered more amenable to one-step distillation due to its "straighter" paths, but this claim has been challenged. Theoretically, optimal score and velocity functions are

interchangeable under Gaussian assumptions. Empirically, Wang et al. (2025) introduce *rectified diffusion*, demonstrating that high-quality noise-image pairs generated by diffusion models perform as well as those produced by flow matching to train ReFlow models. This suggests that the quality of the supervision pairs, rather than the geometry of the sampling path, is the key factor determining the success of ReFlow-based distillation methods. However, these approaches remain fundamentally bounded by the teacher model's generation quality (Wang et al., 2025). In contrast, score distillation has demonstrated the ability to outperform the teacher model, even when using only one sampling step (Zhou et al., 2024; 2025c). Another related line of work is Flow Generator Matching (Huang et al., 2024), which mirrors the derivation of SiD by employing flow-related identities in place of score-based ones. Our unified view of diffusion and flow matching suggests that such reformulations may not always be necessary, as velocity and $x_0$-predictions are linear transformations of each other given the same $x_t$, leading to equivalent training losses used during distillation up to differences in weighting schemes.

## C   WEIGHT NORMALIZED TIME SCHEDULE

We illustrate in Figure 2 the differences between various noise schedules when mapped into the continuous interval $t \in (0, 1)$, assuming an SNR defined as

$$\text{SNR}(t) = \frac{\alpha_t^2}{\sigma_t^2} = \frac{(1-t)^2}{t^2}.$$

The first schedule we consider is the one used by TrigFlow.

The second schedule we consider is the one used by SANA-Sprint.

The third schedule we consider is the one used by SANA, which samples $t \sim \text{logit}\,\mathcal{N}(0, 1)$, but applies a time-step shift to induce a lower SNR compared to the standard rectified-flow schedule at the same $t$. While this schedule still satisfies the identity

$$\alpha_t + \sigma_t = 1,$$

it no longer maintains $\sigma_t = t$. Nonetheless, the resulting distribution of $\sigma_t$ effectively reflects the corresponding distribution of $t$ in rectified flow.

The fourth schedule we consider is the one used by DDPM, for which it is common to apply the $\epsilon$-prediction loss shown in (7), without any additional loss weighting. This is also equivalent to $x_0$-prediction loss shown in (5) weighted by $\text{SNR}(t)$.

The fifth and sixth are the training and inference ones used by EDM.

Comparing the $v$-prediction loss shown in (9) and the $\epsilon$-prediction loss shown in (7), we observe that they differ by a time-dependent scaling factor $\alpha_t^2$. However, as discussed earlier, one must consider both the distribution of $p(t)$ and the weighting function $w(t)$ when evaluating how each $t$ contributes to the overall loss. From this perspective, while the DDPM schedule appears to place more emphasis on values of $t$ closer to one (i.e., by sampling them more frequently), it down-weights the corresponding $x_0$-prediction loss more than the SANA schedule does.

# D ALGORITHMIC PSEUDO-CODE

---

**Algorithm 1** Score Distillation of DiT-Based Flow-Matching T2I generation

---

1: **Input:** Pretrained DiT $v_\phi$, generator DiT $G_\theta$, fake score DiT $v_\psi$, $t_{\text{init}} = 999$, training timestep distribution $p(t) = \text{Logit} \mathcal{N}(t; \mu, \sigma)$, learning rate $\eta$.
2: **Initialization** $\theta \leftarrow \phi, \psi \leftarrow \phi$
3: **repeat**
4:     **Update Fake Score**
5:     Sample $\boldsymbol{z} \sim \mathcal{N}(0, \mathbf{I})$ and $\boldsymbol{x}_g = G_\theta(t_{\text{init}}, \boldsymbol{z})$
6:     Sample $t \sim p(t)$, $\boldsymbol{\epsilon}_t \sim \mathcal{N}(0, \mathbf{I})$ and $\boldsymbol{x}_t = (1 - t)\, \boldsymbol{x}_g + t \boldsymbol{\epsilon}_t$
7:     Use (15) to compute CFG modified $f_\psi(\boldsymbol{x}_t, \boldsymbol{c})$ base on flow prediction $v_\psi(\boldsymbol{x}_t, \boldsymbol{c})$
8:     Update $\psi$ with:
9:         $\mathcal{L}_\psi = \|f_\psi(\boldsymbol{x}_t, \boldsymbol{c}) - \boldsymbol{x}_g\|_2^2, \quad \psi = \psi - \eta \nabla_\psi \mathcal{L}_\psi$
10:     **Update Generator**
11:     Sample $t \sim p(t)$ and compute $\omega_t$ using any combinations listed in the caption of Figure 2; Unless specified otherwise, we used $p(t) = \text{Logit} \mathcal{N}(\ln 2, 1.6^2)$ and $w_t = 1 - t$ for all models in the paper.
12:     Sample generator update step uniformly at random from $k \in \{1, 2, 3, 4\}$. Generate $\boldsymbol{x}_g^{(k)}$ as in (16) and forward diffuse $\boldsymbol{x}_t^{(k)}$ as in (17)
13:     Compute $f_\phi(\boldsymbol{x}_t, \boldsymbol{c})$ base on flow prediction $v_\phi(\boldsymbol{x}_t, \boldsymbol{c})$ following (15)
14:     Compute $f_\psi(\boldsymbol{x}_t, \boldsymbol{c})$ base on flow prediction $v_\psi(\boldsymbol{x}_t, \boldsymbol{c})$ following (15)
15:     Update $G_\theta$ with:
16:         $\mathcal{L}_\theta(\boldsymbol{x}_t^{(k)}) = w_t \big(f_\phi(\boldsymbol{x}_t^{(k)}, t_k, c) - f_\psi(\boldsymbol{x}_t^{(k)}, t_k, \boldsymbol{c})\big)^T \big(f_\psi(\boldsymbol{x}_t^{(k)}, t_k, \boldsymbol{c}) - \boldsymbol{x}_g^{(k)}\big)$
17:         $\theta = \theta - \eta \nabla_\theta \mathcal{L}_\theta$
18: **until** the FID plateaus or the training budget is exhausted
19: **Output:** $G_\theta$

---

# E DETAILED GENEVAL SCORES

Table 3: GenEval scores and per-task accuracies (single_object, counting, color_attr, colors, position, two_object) for SANA, SANA Sprint, SiD-DiT, and $\text{SiD}_\alpha^2$-DiT across 0.6B and 1.6B backbones.

| Model | GenEval | single_object | counting | color_attr | colors | position | two_object |
|---|---|---|---|---|---|---|---|
| **SANA 0.6B** | | | | | | | |
| SANA (Xie et al., 2025) | 0.64087 | 1.0000 | 0.6281 | 0.4225 | 0.8910 | 0.2044 | 0.6992 |
| SiD-DiT (SANA) | 0.65212 | 0.9812 | 0.5969 | 0.4475 | 0.8484 | 0.2800 | 0.7727 |
| $\text{SiD}_\alpha^2$-DiT (SANA) | 0.64307 | 0.9812 | 0.6312 | 0.4375 | 0.8617 | 0.2700 | 0.7626 |
| **SANA 1.6B** | | | | | | | |
| SANA (Xie et al., 2025) | 0.65501 | 0.9969 | 0.5782 | 0.3925 | 0.8856 | 0.2770 | 0.7998 |
| SiD-DiT (SANA) | 0.67013 | 0.9781 | 0.5437 | 0.4950 | 0.8590 | 0.3000 | 0.8333 |
| $\text{SiD}_\alpha^2$-DiT (SANA) | 0.66472 | 0.9812 | 0.6219 | 0.4450 | 0.8484 | 0.2425 | 0.7601 |
| **SANA TrigFlow 0.6B** | | | | | | | |
| SANA Sprint Teacher | 0.78018 | 1.0000 | 0.7000 | 0.5575 | 0.9069 | 0.5975 | 0.9192 |
| SANA Sprint (TrigFlow, 1 step) | 0.77074 | 0.9938 | 0.6469 | 0.4650 | 0.8883 | 0.5300 | 0.8005 |
| SANA Sprint (TrigFlow, 4 steps) | 0.76591 | 1.0000 | 0.6812 | 0.5150 | 0.8803 | 0.6300 | 0.8889 |
| SiD-DiT (SANA, TrigFlow) | 0.76289 | 1.0000 | 0.5594 | 0.5050 | 0.8936 | 0.5700 | 0.9116 |
| $\text{SiD}_\alpha^2$-DiT (SANA, TrigFlow) | 0.77251 | 0.9906 | 0.6938 | 0.4875 | 0.8803 | 0.6275 | 0.9141 |
| **SANA TrigFlow 1.6B** | | | | | | | |
| SANA Sprint Teacher | 0.77571 | 1.0000 | 0.6719 | 0.5725 | 0.8830 | 0.5875 | 0.9394 |
| SANA Sprint (TrigFlow, 1 step) | 0.76796 | 0.9938 | 0.5219 | 0.5025 | 0.8936 | 0.5425 | 0.8535 |
| SANA Sprint (TrigFlow, 4 steps) | 0.76769 | 1.0000 | 0.5844 | 0.5275 | 0.9149 | 0.5525 | 0.8889 |
| SiD-DiT (SANA, TrigFlow) | 0.77421 | 0.9875 | 0.6219 | 0.5350 | 0.9096 | 0.6350 | 0.9369 |
| $\text{SiD}^2$-DiT (SANA, TrigFlow) | 0.76829 | 0.9906 | 0.6562 | 0.5175 | 0.8617 | 0.5775 | 0.9015 |

# F HYPERPARAMETER SETTINGS

Table 4: Comparison of distillation time and memory usage for training four-step generators from SANA Rectified Flow (0.6B or 1.6B) or SANA TrigFlow teachers. Measurements exclude the overhead of text encoding.

| Computing platform | Hyperparameters | 0.6B 512x512 | 1.6B 512x512 | 0.6B 1024x1024 | 1.6B 1024x1024 |
|---|---|---|---|---|---|
| | Teacher Model | Rectified Flow | Rectified Flow | TrigFlow | TrigFlow |
| General Settings | # of learnable parameters (fp32 model size in GB) | | | | |
| | VAE Size (fp32 model size in GB) | | | | |
| | Text Encoder Size (fp32 model size in GB) | | | | |
| | Learning rate | | | 5e-6 | |
| | Optimizer | | Adam ($\beta_1 = 0, \beta_2 = 0.999, \epsilon = $ 1e-8) | | |
| | $\alpha$ | | | 1 | |
| | $\lambda_{\text{sid}}$ | | | 100 | |
| | # of GPUs | | | 8xH100 (80G) | |
| | Batch size | | | 256 | |
| | VAE offload to CPU | | | Yes | |
| SiD-DiT (4 steps) | Batch size per GPU | 16 | 16 | 8 | 4 |
| AMP+FSDP | # of gradient accumulation round | 2 | 2 | 4 | 8 |
| | Max memory in GB allocated | 38 | 65 | 66 | 71 |
| | Max memory in GB reserved | 42 | 74 | 70 | 77 |
| | Time in seconds per 1k images | 16 | 19 | 49 | 108 |
| | Time in hours per 1M images | 5 | 5 | 14 | 30 |

Table 5: Comparison of distillation time and memory usage for training four-step generators from four teacher models: SD3-Medium, SD3.5-Medium, SD3.5-Large, and FLUX.1-dev (under both 512x512 and 1024x1024 resolutions). We evaluate two methods: four-step SiD-DiT, a data-free approach that requires no real images, and four-step SiD$_2^a$-DiT, which initializes from a SiD-DiT-distilled generator and continues training with an additional Diffusion-GAN-based adversarial loss using user-provided data. Measurements exclude the overhead of text encoding in SiD and both text and image encoding in SiD$_2^a$, which can be either precomputed or batch-processed outside the main distillation loop; the latter strategy is used in this work.

| Computing Platform | Method | SD3-Medium | SD3.5-Medium | SD3.5-Large | FLUX.1-dev | FLUX.1-dev |
|---|---|---|---|---|---|---|
| | Resolution | 1024x1024 | 1024x1024 | 1024x1024 | 512x512 | 1024x1024 |
| | # of learnable parameters (fp32 model size in GB) | | | | | |
| | VAE Size (fp32 model size in GB) | | | | | |
| | Text Encoder Size (fp32 model size in GB) | | | | | |
| General Settings | $\alpha$ | | | 1 | | |
| | $\lambda_{\text{sid}}$ | | | 100 | | |
| | # of GPUs | 8xH100 (80G) | 8xH100 (80G) | 8xH100 (80G) | 8xH100 (80G) | 8xB200 (192G) |
| | Batch Size | | | 256 | | |
| | Learning Rate | | | 1e-6 | | |
| | Optimizer | | Adam ($\beta_1 = 0, \beta_2 = 0.999, \epsilon = $ 1e-8) | | | |
| | Gradient Clipping | | | No | | |
| | CPU Offloading | No | No | Yes | – | – |
| SiD-DiT (4 steps) | Batch Size per GPU | 2 | 2 | 1 | – | – |
| AMP+FSDP | # of Gradient Accumulation Rounds | 16 | 16 | 32 | – | – |
| | AMP + FSDP: Max Memory Allocated (GB) | 57 | 62 | 72 | – | – |
| | AMP + FSDP: Max Memory Reserved (GB) | 67 | 73 | 77 | – | – |
| | Time per 1k Images (s) | 150 | 230 | 1000 | – | – |
| | Time per 1M Images (h) | 42 | 64 | 277 | – | – |
| | Learning Rate | | | 1e-5 | | |
| | Optimizer | | Adam ($\beta_1 = 0, \beta_2 = 0.999, \epsilon = $ 1e-4) | | | |
| | Gradient Clipping | | | Yes | | |
| | CPU Offloading | | | No | | |
| SiD-DiT (4 steps) | Batch Size per GPU | 4 | 4 | 1 | 1 | 2 |
| BF16+FSDP | # of Gradient Accumulation Rounds | 8 | 8 | 32 | 32 | 16 |
| | AMP + FSDP: Max Memory Allocated (GB) | 47 | 69 | 56 | 60 | 146 |
| | AMP + FSDP: Max Memory Reserved (GB) | 55 | 78 | 70 | 74 | 165 |
| | Time per 1k Images (s) | 120 | 200 | 550 | 650 | 720 |
| | Time per 1M Images (h) | 33 | 56 | 153 | 181 | 200 |
| | Learning Rate | | | 1e-6 | | |
| | Optimizer | | Adam ($\beta_1 = 0, \beta_2 = 0.999, \epsilon = $ 1e-8) | | | |
| | Gradient Clipping | | | No | | |
| | CPU Offloading | No | No | Yes | – | – |
| SiD$_2^a$-DiT (4 steps) | Batch Size per GPU | 4 | 4 | 1 | – | – |
| BF16+FSDP | # of Gradient Accumulation Rounds | 8 | 8 | 32 | – | – |
| | AMP + FSDP: Max Memory Allocated (GB) | 47 | 69 | 62 | | |
| | AMP + FSDP: Max Memory Reserved (GB) | 56 | 78 | 73 | | |
| | Time per 1k Images (s) | 138 | 240 | 670 | – | – |
| | Time per 1M Images (h) | 38 | 67 | 186 | – | – |

Table 6: Estimated training cost of SiD-DiT with different teacher models, measured in thousands of images processed (*k imgs*) and in estimated machine hours, shown for both training to the final checkpoint and for reaching near-converged metrics. All estimates are based on a single node with eight H100 GPUs, except for FLUX 1024 Res, which used eight B200 GPUs. The near-converged points are inferred from Figure 4. Estimated training times (in hours) are computed as the number of images iterated (in millions) multiplied by the time-per-million-images values reported in Tables 4 and 5.

| Model | k imgs to checkpoint | k imgs to near convergence | Hours to checkpoint | Hours to near convergence |
|---|---|---|---|---|
| SANA 0.6B (512 res) | 665 | 100 | 3.325 | 0.5 |
| SANA 1.6B (512 res) | 1996 | 400 | 9.98 | 2 |
| SANA 0.6B (1024 res, TrigFlow) | 1587 | 200 | 22.218 | 2.8 |
| SANA 1.6B (1024 res, TrigFlow) | 1638 | 150 | 49.14 | 4.5 |
| SD3 Medium | 1669 | 200 | 55.077 | 6.6 |
| SD3.5 Medium | 1269 | 400 | 71.064 | 22.4 |
| SD3.5 Large | 696 | 130 | 106.488 | 19.89 |
| Flux 512 Res | 778 | 778 | 140.818 | 140.818 |
| Flux 1024 Res | 699 | 300 | 139.8 | 60 |

## G ADDITIONAL QUALITATIVE EXAMPLES

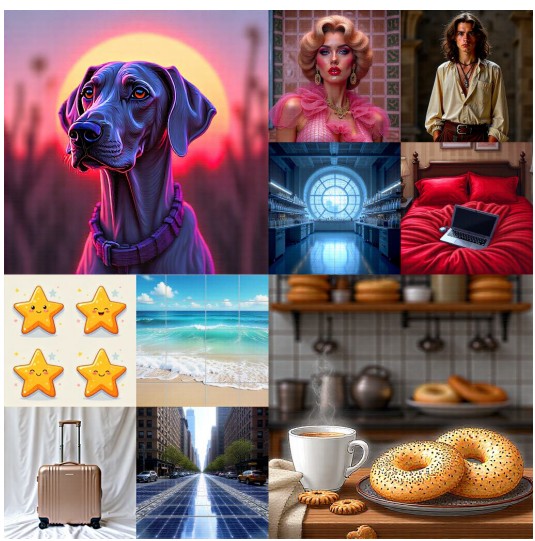

Figure 5: Qualitative results produced by the four-step SiD-DiT generator distilled from FLUX-1.DEV.

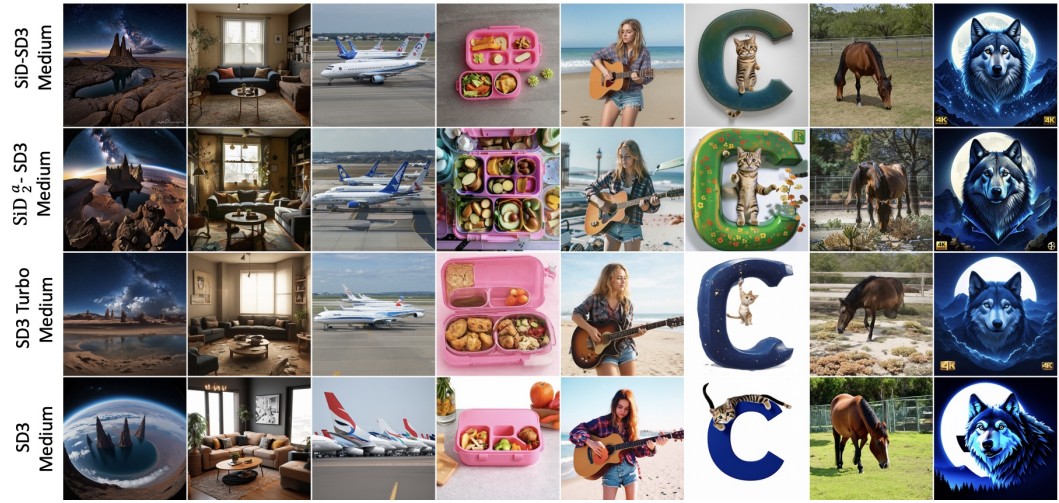

Figure 6: Qualitative results from the four-step SiD-DiT, SiD$_2^a$-DiT, Flash Diffusion SD3, and the teacher model SD3-MEDIUM.

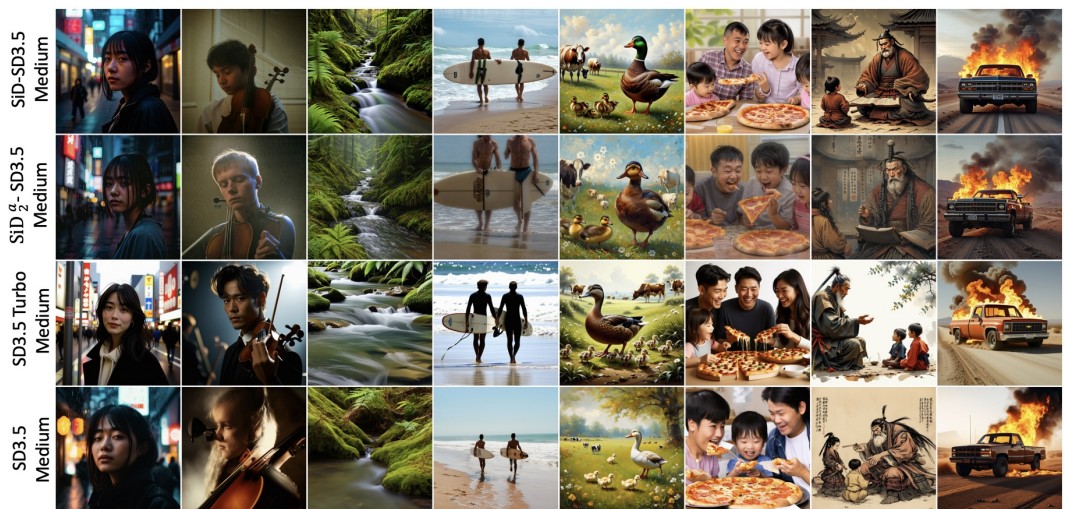

Figure 7: Qualitative results from the four-step SiD-DiT and SiD$_2^a$-DiT generators distilled from SD3.5-MEDIUM, compared against SD3.5-TURBO-MEDIUM and the teacher model SD3.5-MEDIUM.

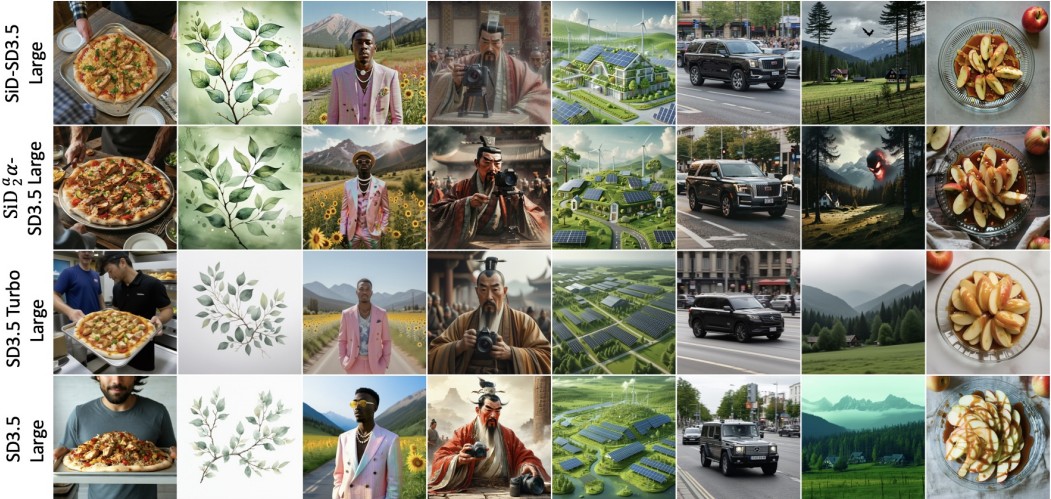

Figure 8: Qualitative results from the four-step SiD-DiT and SiD$_2^a$-DiT generators distilled from SD3.5-LARGE, compared against SD3.5-TURBO-LARGE and the teacher SD3.5-LARGE.

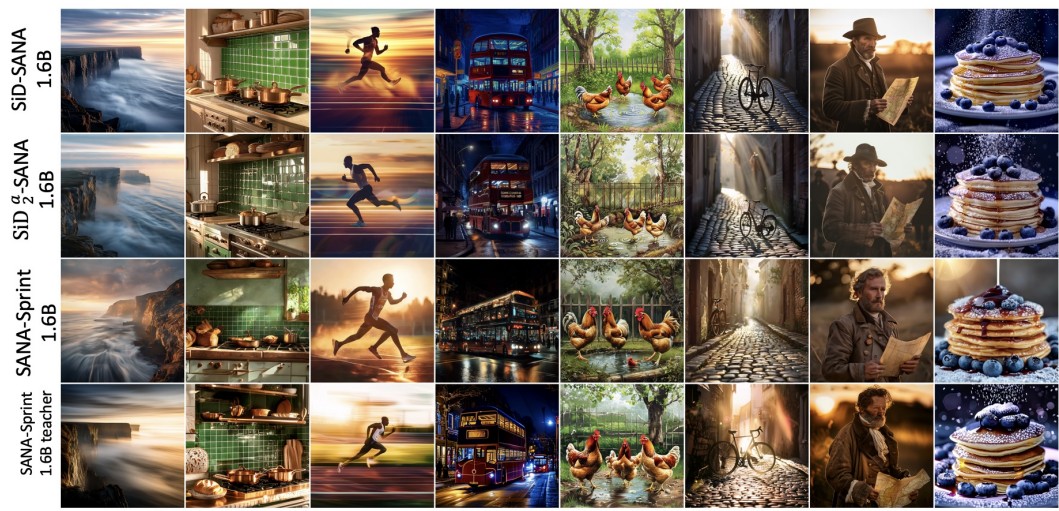

Figure 9: Qualitative results from the four-step SiD-DiT and SiD$_2^a$-DiT generators distilled from the SANA-SPRINT teacher (1.6B), compared against SANA-SPRINT 1.6B and the teacher.

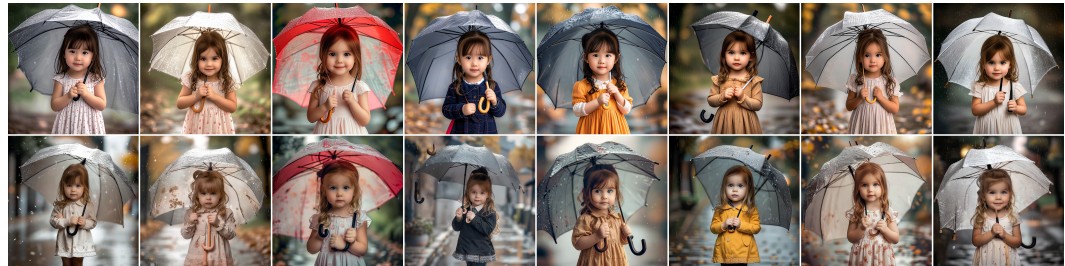

Figure 10: Qualitative results from the four-step SiD-DiT (row 1) and SiD$_2^a$-DiT (row 2) generators distilled from the SANA-SPRINT teacher (1.6B). The prompt used for generation is: "A little girl is posing for a picture and holding an umbrella."

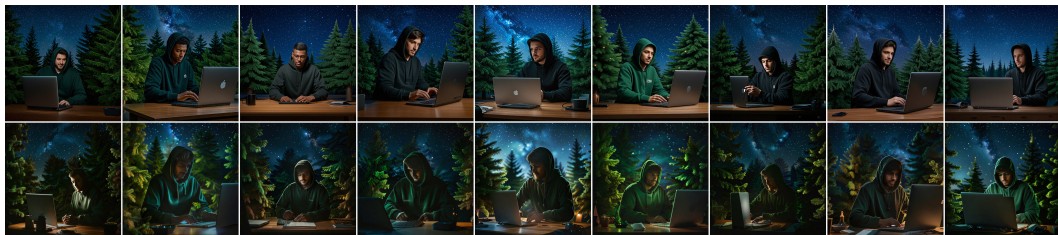

Figure 11: Qualitative results from the four-step SiD-DiT (row 1) and SiD$_2^a$-DiT (row 2) generators distilled from the SANA-SPRINT teacher (1.6B). The prompt used for generation is: "stars in the night sky, majestic green forest trees, guy in a hoodie at a computer."

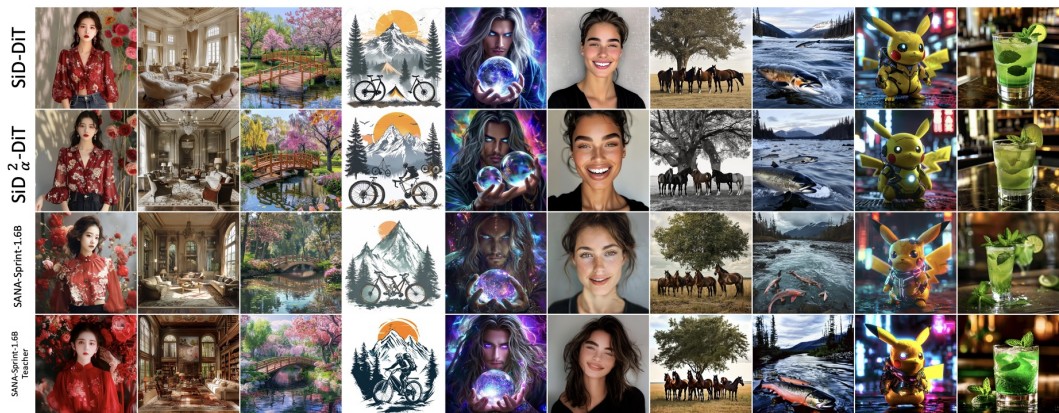

Figure 12: Additional Qualitative results from the four-step SiD-DiT and SiD$_2^a$-DiT generators distilled from the SANA-SPRINT teacher (1.6B). using prompts for generating Figure 1.

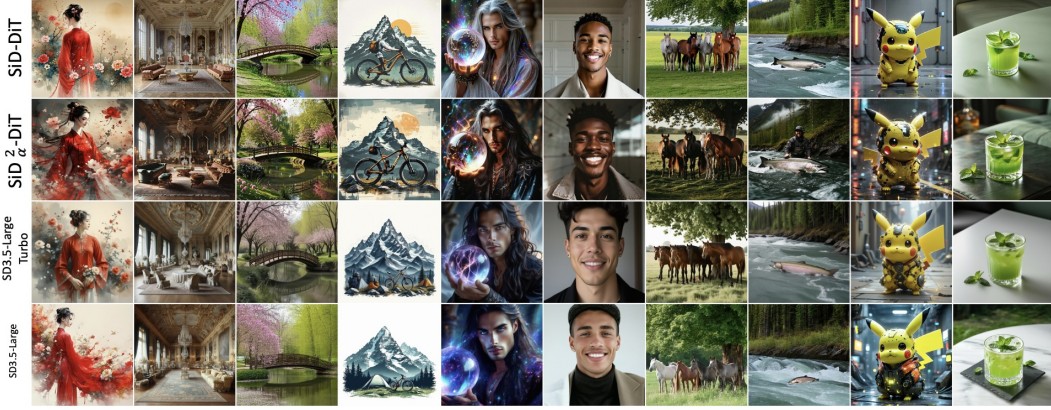

Figure 13: Additional Qualitative results from the four-step SiD-DiT and SiD$_2^a$-DiT generators distilled from SD3.5-LARGE, compared against SD3.5-TURBO-LARGE and the teacher model SD3.5-LARGE using prompts for generating Figure 1.

## H  PROMPT DETAILS

Prompts used for generating Figure 1:

1. chinese red blouse, in the style of dreamy and romantic compositions, floral explosions –ar 24:37 –stylize 750 –v 6

2. "A large room with furniture in the style of Ludwig 14. "

3. "a park with a beautiful wooden bridge over a pond, flowering trees around the banks, beautiful color correction, 16 k, pastel "

4. "design graphic mountain ,camping and bike , white background, no mockup "

5. "beautiful man with long hair and silver eyes holding a huge ornate crystal ball, magical, electric, vivid colors"

6. "Portrait of a instagram model, face facing straight towards the camera, looking into the camera, man, smiling, chic modernist style, unsplach, I cant believe how beautiful this is ",

7. "a group of horses standing next to a tree in an open field"

8. "river in alaska with salmon"

9. "pikachu from the future, Cyberpunk, TRON, 8k, octane render, hyper realistic, photo realistic "

10. "a cocktial made of a green herbal liqueur with fresh peppermint, nice lounge athomsphere, real photo"

Prompts used for generating Figure 5:

1. "Weimaraner synthwave, 80s sunset in background",

2. "james bidgood style image of hollywood female ingenue of the year 1982 ",

3. "27 year old man, with necklength brown wavy hair, in medieval shirt and trousers, fantasy, dramatic lighting, 169",

4. "panorama photography shot of a science lab bright light in window",

5. "A bed with red sheets on it and messy blanket and a lap top.",

6. "star badges for children, similar style but different variations, flat illustration, cute, dribble, behance, very cute, happy star ",

7. "Clear distinct beach waves pattern HD tile caribbean1",

8. "Two small suitcase is sitting in front of a white sheet.",

9. "Manhattan streets paved with glossy solar panels",

10. "A counter filled with coffee, cookies, and bagels.",

Prompts used for generating Figure 6:

1. "High altitude photo of a planet, cloud later, tall peaked towers surrounded by water reflecting starlight, and rocky deserts. Fisheye lens. Milkyway background. ",

2. "afroamerican household, hiphop themed living room, a bit messy, high resolution, 4k, 5 v",

3. "Modern jet airplanes lined up on the runway ready for take off",

4. "Pink lunch box with compartments for all types of food",

5. "young woman playing the guitar on Venice Beach in 1994, shes wearing denim shorts and a flannel, In the style of Petra Collins, 90s, grunge fashion, pastel coloring, cinematic color grading. ",

6. "a cat climbing up a LARGE, letter C, pixar, white background ",

7. "The horse is grazing in the fenced coral.",

8. "a logo of wolf, blue light shadow, ultra realistic, 4k hd, full moon, mountains ",

Prompts used for generating Figure 7:

1. "some kind of chicken, rice, and vegetable dish on a pizza tray being served to a man.",

2. "a dainty watercolor twig with leaves in sage green, on white background, simplistic",

3. "Portrait of man wearing pastel colored fancy suit, tyler the creator inspired, round bead jewlery necklace, sun flower field mountain with a road in between the mountatins. Photo is taken with a 12mm f1.2 canon lens",

4. "a hyper realistic image of Confucious speaking on the camera in ancient times ",

5. "renewable energy, green, sustainable, ecology, community, 3d, concept art, long shot",

6. "The large SUV drives along a busy street.",

7. "serene countryside vista with detail of homes, forest, mountains with something evil lurking amongst the trees hidden in shadows, 8k v5 ",

8. "A glass plate topped with sliced apples and caramel. ",

Prompts used for generating Figure 8:

1. Street portrait in Shibuya at dusk, shallow DOF, neon bokeh, light rain on pavement, candid framing",

2. "Editorial portrait of a violinist backstage, tungsten rim light, light haze, shallow DOF, subtle grain",

3. "Mossy canyon stream, slow shutter silky water, fern details, cool color grade",

4. "Two surfers walk down the beach holding their boards.",

5. "A hyper detail painting in richard macneil style of a duck with her ducklings, walking through a field were there are cows grazing ",

6. "An Asian family that is eating pizza together.",

7. "old samurai telling stories to his children",

8. "car magazine advertising photography, 80s pickup truck, engulfed in flames, high noon, apocalyptic desert, empty road, cinematic composition and lighting, cinematic photography. ",

