# OpenReview forum: "Score Distillation of Flow Matching Models"
_ICLR.cc/2026/Conference — Submitted to ICLR 2026_

### Official Review · Reviewer_iCKc · 2025-10-17

**Soundness:** 3
**Presentation:** 3
**Contribution:** 3
**Rating:** 6
**Confidence:** 3

**Summary:**

The work proposed to extend Score identity Distillation (SiD) framework to distill flow-based text-to-image models into few step generators, SiD-DiT. The key contributions of the work include the analysis of loss reweighting for pretraining of diffusion and flow-matching models, which show the equivalence of optimized objectives with the only difference of timestep weighting functions, and extension of the SiD to distill pretrained flow-based text-to-image models SANA, SD3-Medium, SD3.5-Medium/Large, and FLUX.1-Dev. Distilled models were evaluated on 10k COCO-2014 validation subset with FID and CLIP, and on GenEval benchmark. The proposed distilled models were compared with teacher models, and distilled models of SANA Sprint, Flash SD3, SD3.5-Turbo-Medium, and SD3.5-Turbo-Large. The results show that SiD-DiT training framework can improve the quality of the teacher model in terms of FID and CLIP for most flow-based text-to-image models, while having only 4 NFE.

**Strengths:**

1) Experimental results of the proposed SiD-DiT model on different flow-based text-to-image models, including SANA, SANA TrigFlow, SD3, SD3.5, and FLUX with different model sizes support the claim of the work that SiD can be applicable to flow-based models to maintain high-quality image results of teacher models with big NFE, while having much faster inference with NFE=4 only.
2) The unified view presented in Section 2.3 on diffusion and flow-based objectives supports the motivation of the work to apply SiD for flow-based text-to-image models.
3) The distillation of the FLUX model with 12B parameters into 4 step generator requires complex technical implementation to achieve it on a single node with 8 GPUs. In my opinion, this result is impressive since there almost no models to distill such a large text-to-image model.
4) In contrast to the recent SANA-Sprint model, which is based on consistency distillation and requires real data, the proposed SiD-DiT enables data-free distillation for flow-based text-to-image models. Table 1 in the paper shows that even without real data SiD-DiT achieves competitive results when compared with the SANA-Sprint model.

**Weaknesses:**

1) In contrast to the original SiD paper, which applied the SiD objective to distill diffusion model in 1-step generator, SiD-DiT does not explore the option NFE=1. It makes the comparison of SiD-DiT with SANA Sprint, which has NFE=1 for SANA TrigFlow 1.6B model, to be unfair. The DMD2 paper motivated the multi-step generator found that the distillation of SDXL model into 1-step generator is challenging, as shown by Patch FID in Table 2 of DMD2. The work of Zhou et al. for multistep SiD with diffusion SDXL model in Table 6 shows the improvements of multistep SiD model with NFE=4 over NFE=1 in terms of FID, but also the drop in CLIP. It remains a question on the quality of SiD-DiT with NFE=1 or NFE=2.
2) The work lacks of discussion on the stability of the SiD framework for flow-based text-to-image models, which was done in prior distillation works of Zhou et al. (Figure 9) and DMD2 (Figure 9) for diffusion text-to-image models. This is important to understand why SiD-DiT for FLUX.1-Dev achieved much worse results compared to the teacher model: is it due to computational limitations in terms of training time and the proposed method did not converge or the distillation converged, but the results are worse due to unique features of FLUX.1-Dev such as learned guidance embedding.
3) The work does not provide detailed pseudo-code of the proposed method, which is crucial to articulate the difference and similarities between the details and heuristics of the SiD-DiT implementation for flow-based text-to-image models with the Algorithm 1 in Zhou et al. of SiD for diffusion text-to-image models and Algorithm 1 in the original SiD paper.
4) Even though the SiD-DiT method is compared with teacher flow-based models and several distillation models of these models, the work lacks of comparison with existing distilled few-step diffusion-based models in terms of image quality and inference efficiency trade-off.

References:

Zhou et al. Few-Step Diffusion via Score identity Distillation, arxiv-2025.

DMD2. Improved Distribution Matching Distillation for Fast Image Synthesis, NeurIPS-2024.

**Questions:**

1) In the original SiD paper, the authors found that $\alpha \neq 1$ can slightly improve the results (see Figure 2 and Figure 8 in the SiD paper). Did you try to vary the $\alpha$ parameter?
2) Since the results of SiD-DiT and SANA Sprint models are closed in the mean GenEval score, can you proved detailed 6 GenEval scores for SANA, SANA Sprint, and SiD-DiT models?
3) As far as I understood, the whole Section 2 is given only to motivate the application of SiD for flow-based models. Are there any novel practical implications or techniques, which were used to implement SiD-DiT in Section 3 based on the theory from Section 2?
4) Can you comment on the ability SiD-DiT to distill flow-based text-to-image models in 1 or 2 steps?
5) Can you comment on the stability of SiD-DiT distillation training for all experiments? Did the distillation converged for the FLUX.1-Dev experiment?
6) Can you comment on the comparison in terms of image quality and inference efficiency trade-off for SiD-DiT and other recent fast text-to-image models, including DMD2, FLUX-schnell, and Rectified Diffusion (Phased)?
7) It should be noted that FID and CLIP metrics have issues with correspondence of human preferences in outputs of text-to-image models, as noted by PickScore and ImageReward papers. Can you provide PickScore and ImageReward metrics for SiD-DiT and other models from Table 2?

References:

Rectified Diffusion: Straightness Is Not Your Need in Rectified Flow. ICLR-2025.

Pick-a-Pic: An Open Dataset of User Preferences for Text-to-Image Generation. NeurIPS-2023.

ImageReward: Learning and Evaluating Human Preferences for Text-to-Image Generation. NeurIPS-2023.

---

> ### Author Response · Authors · 2025-11-24
>
> We thank Reviewer **iCKc** for the positive feedback and thoughtful comments. We appreciate the recognition of our work’s strengths: including the unified theoretical view, the broad applicability of SiD-DiT across diverse flow-based T2I models, the ability to FLUX-1-Dev on a single node, and the advantages of data-free distillation. Below, we address the remaining concerns point-by-point.
>
> **Weakness 1: Comparison at NFE = 1**
>
>    - As shown in the **updated Table 1**, SiD consistently outperforms the 4-step SANA-Sprint baseline, achieving lower FID and higher CLIP scores for both the 0.6B and 1.6B models. Please also refer to our response to Question 4.
>
> **Weakness 2: FLUX-1-Dev**
>
>   - We note that our experiments did not incorporate FLUX-specific adaptations, such as adjustments to the learned guidance embedding or modifications tailored to the FLUX network configuration. These components may require additional changes. Unfortunately, due to limited computational resources, we were unable to perform a full ablation or investigate these adaptations in depth.
>
>   - Nonetheless, our results demonstrate that **SiD-DiT is compatible with the FLUX architecture**, and we view incorporating real data to further enhance distillation, along with comprehensive tuning and a deeper investigation of the optimal training dynamics for FLUX-1-Dev, as valuable directions for future work.
>
> **Weakness 3: Lack of Psedo code**
>
>    - Thanks for pointing this out. We have added pseudo-code into Appendix D.
>
> **Weakness 4: Comparison with Few-Step Diffusion Baselines**
>    - Our goal is to propose a *single, general-purpose* distillation framework for flow-based T2I models. Therefore, we prioritize **within-family comparisons** to avoid confounding architectural differences.
>
>   - Additionally, most publicly available distilled diffusion models operate at **much smaller scales** and are not directly comparable to modern flow-based T2I models like SANA, SD3.5, or FLUX. Cross-family comparisons could therefore be misleading rather than informative.

---

> ### Author Response · Authors · 2025-11-24
>
> **Question 1: Effect of the $\alpha$ Parameter**
>
>  - SiD-DiT is designed to serve as an out-of-the-box distillation pipeline for a wide range of open-source flow-based models. Given the diversity of architectures and noise schedules, we did not tune $\alpha$. Instead, we adopt $\alpha = 1$, which has consistently demonstrated stable behavior in practice, even though it may not be the optimal choice for a specific dataset or model.
>
> **Question 2: Detailed GenEval Scores**
>
>  - We haved added the detailed 6 GenEval scores to **Table 3** in the Appendix, including results for SANA, SANA-Sprint, and SiD-DiT.
>
> **Question 3: Practical Use of Theory in Section 2**
>
>  - The theoretical results in Section 2 provide a unified **score-based formulation** for flow-matching models. This enables us to:
>     - treat flow objectives analogously to diffusion objectives,
>     - directly incorporate flow-based noise schedules, and
>     - apply a streamlined SiD pipeline across diverse flow-pretrained models.
>
>   - These theoretical insights directly inform the **design principles in Section 3**, guiding loss design, reweighting choices, and timestep parameterization.
>
> **Question 4: Distilling to NFE = 1 or 2**
>
>   - In our preliminary experiments, 1-step SiD-DiT models achieve FID and CLIP scores comparable to their multi-step variants, but 4-step distillation provides clearly better fine-grained texture and spatial coherence. Moreover, because text encoding and VAE decoding constitute a large portion of the inference cost, increasing the number of steps from 1 to 4 leads to only a modest increase in total runtime (approximately **1.5×** the cost of a single forward pass).
>
>    - For these reasons, we adopt **4-step distillation as the best quality–efficiency trade-off**.
>
>
> **Question 5: Stability and Convergence Across Experiments**
>
>   - As discussed under Weakness 2, we were unable to perform a full convergence and ablation study for FLUX-1-Dev due to resource constraints. Across all other models (SANA 0.6B/1.6B/Flow/TrigFlow, SD3, SD3.5 Medium/Large), **SiD-DiT training was stable and converged reliably**.
>
>    - We consider further investigation into optimal FLUX adaptations an important direction for future work.
>
> **Question 6: Comparison with Fast Diffusion Models**
>
>    - Direct comparisons across architectures (e.g., Distilled Diffusion vs. Distilled Flow models) are often misleading because quantitative metrics do not capture inherent architectural differences.
>
>    -  For example, SANA achieves higher GenEval scores than FLUX, yet FLUX produces stronger visual quality.
>
>    -  We have updated **Table 2** to include **FLUX-Schnell** results, enabling a partial quality–efficiency comparison.
>
>    - Rectified Diffusion (Phased) has not been tested for any of our base models, and SenseFlow’s DMD2 results for SD3.5 and FLUX are unavailable, preventing controlled comparisons.
>
> **Question 7: Human-Preference Metrics**
>
>   - We have added **PickScore**, **ImageReward**, **LAION-Aesthetics**, and **HPSv2** to both Table 1 and Table 2.
> These results show that SiD-DiT maintains consistently competitive, and often *better*, alignment with human preference compared to teacher models and adversarial/consistency-based baselines.
>
>   - Notably, on **SD3.5-Large**, SiD-DiT surpasses the teacher in **3 out of 4** human-preference metrics, demonstrating that our efficiency improvements do not necessarily come at the cost of perceptual quality.
>
> We hope these clarifications address your questions and strengthen the presentation of our contributions.

---

### Official Review · Reviewer_mxjm · 2025-10-31

**Soundness:** 3
**Presentation:** 3
**Contribution:** 2
**Rating:** 4
**Confidence:** 4

**Summary:**

This paper presents a unified theoretical and practical framework for score distillation applied to flow matching models. The authors first show that, under Gaussian noise assumptions, diffusion and flow matching formulations are mathematically equivalent up to a reweighted time distribution. Building on this equivalence, they adapt the Score Identity Distillation method—originally designed for diffusion models—to DiT-based flow matching architectures, forming SiD-DiT. The proposed approach enables few-step (e.g., four-step) sampling without requiring teacher fine-tuning or architectural modifications. Extensive experiments on large-scale text-to-image models (SANA, SD3.5, FLUX) demonstrate that SiD-DiT achieves comparable or superior performance to specialized fast samplers (e.g., SANA-Sprint, SD3.5-Turbo) while maintaining generality across architectures.

**Strengths:**

1. The paper provides a clear and rigorous theoretical unification between diffusion and flow matching models via Bayes conditioning and Tweedie’s formula, avoiding the need for SDE/ODE derivations.
2. The proposed SiD-DiT framework is architecturally and algorithmically minimal—it directly applies score distillation to flow matching models without teacher or network modification.
3. Empirical validation is comprehensive, covering multiple model families (SANA, SD3.5, FLUX) and showing that 4-step generation matches or surpasses teacher quality.

**Weaknesses:**

1. The novelty is more integrative than conceptual—theoretical equivalence between diffusion and flow matching is known, though not previously formalized this elegantly.
2. The adversarial extension is underexplored—it improves FID but lacks theoretical grounding or stability discussion.
3. The paper could benefit from more detailed comparisons to existing distillation frameworks like Progressive Distillation, Consistency Models, or MeanFlow, especially regarding training cost and convergence dynamics.

**Questions:**

1. How sensitive is SiD-DiT to the specific choice of timestep distribution $p(t)$ and weighting $w(t)$ ? Could these be learned adaptively?
2. In the large-model experiments (e.g., FLUX-1), is there evidence that student models inherit biases or artifacts from the teacher’s flow parameterization?

---

> ### Author Response · Authors · 2025-11-24
>
> Thank you for the valuable feedback and for recognizing the strengths of our work, including the clarity of our theoretical unification, the minimal and architecture-agnostic design of SiD-DiT, and the comprehensive empirical validation across multiple DiT-based flow-matching models. Below, we address the remaining concerns point-by-point.
>
> **1. Scope of Theoretical Contribution**
>
>    * While prior work has noted limited connections between diffusion and flow matching, our theoretical results in **Section 2** go substantially further by developing a comprehensive *score-distillation formulation* tailored to modern flow-matching T2I models. Specifically:
>
>        - We show that a *broad class of flow objectives* share the same optimal solution under our framework.
>        - We demonstrate how *flow-based noise schedules* (including TrigFlow) integrate seamlessly into this unified formulation.
>        - Building on this score-based view, we derive *practical design principles*, including loss formulations, timestep parameterizations, and reweighting schemes, that enable a *single SiD-DiT pipeline* to operate across heterogeneous DiT-based flow architectures (Efficient-DiT, MMDiT, DC-AE variants) and diverse pretraining objectives.
>
>   * Together, these results provide both a strengthened theoretical foundation and a unified practical instantiation, representing a substantive conceptual and methodological advance beyond a purely integrative unification.
>
> **2. Adversarial Extension**
>
>  - The adversarial component plays an **auxiliary role** in our framework and is found to be robust in practice, consistently improving FID for both the SANA and SD3 model families. It is most effective when real data is available, as it enhances generative sharpness and provides a strong regularization effect. In our current experiments, its impact is limited by the exclusive use of synthetic data, since we did not have access to real datasets at the scale typically required for training modern DiT base models.
>
>   - Importantly, our method already performs **very strongly in a data-free setting**, which distinguishes it from existing distillation approaches for DiT models that typically rely on access to high-quality real data. Introducing the adversarial component makes our method partially data-dependent, but only to a limited extent, as the adversarial term serves purely as an auxiliary mechanism to help the student generator better adapt to the available data.
>
> **3. More Comparison to Distillation Baselines**
>
>  * Thank you for the suggestions.
>
> * **Progressive Distillation** is not in conflict with our framework. In principle, one could first distill a 16-step generator, then progressively reduce it to 8, 4, 2, or even 1 step. Our method focuses on unifying score distillation across flow-matching models, and progressive reduction could be integrated on top of SiD-DiT if desired.
>
> * We note that **SANA-Sprint**, a recent method trained via a state-of-the-art **consistency distillation** approach, is already included in our evaluation.
>
> * **MeanFlow** differs fundamentally from our setting: its main goal is to train a one- or few-step generator *from scratch*, rather than following the pretrain-then-distill pipeline used in modern T2I systems.
>   - On standard non-T2I benchmarks (e.g., CIFAR-10, ImageNet at various resolutions), its performance still falls noticeably behind state-of-the-art pretrain-plus-distill methods such as SiD and SiDA.
>    - Moreover, at the time of submission, MeanFlow has only been demonstrated at relatively small scales and **has not been validated on modern T2I architectures**, which are significantly more challenging in both robustness and scalability.
>    - In other words, to the best of our knowledge, there are currently **no MeanFlow-based T2I baselines** suitable for a controlled comparison.

---

> ### Author Response · Authors · 2025-11-24
>
> **Question 1: On the Selection of $p(t)$ and $w(t)$**
>
>   - While iterative-refinement–based ODE/SDE solvers are often sensitive to the choices of $p(t)$ and $w(t)$, our theory shows that the *optimal solutions of the score-identities are independent* of these choices. This theoretical invariance directly contributes to the strong empirical robustness of SiD-DiT across different timestep distributions and weighting schemes.
>
>  - Our empirical observation is that **as long as $p(t)$ and $w(t)$ together provide sufficient coverage across the noise spectrum**, SiD-DiT performs consistently well. Please refer to **Section 4.1** for more detailed discussion of our design choices for $p(t)$ and $w(t)$.
>
>  - Learning $p(t)$ and $w(t)$ is an interesting direction. To our knowledge, there is no prior work exploring adaptive or learned timestep distributions for distillation in flow-matching models. We believe this could be particularly useful if RL-based objectives are introduced—for example, emphasizing perceptual vividness, which might benefit from allocating more timesteps to high-noise regions.
>
> **Question 2 — Inheritance of Teacher Biases**
>
>   - Vanilla score distillation encourages the student model to match the teacher’s distribution; therefore, SiD-DiT naturally inherits any biases or artifacts present in the teacher.
>   - However, recent work on **Score Forgetting Distillation (SFD)** [1] shows that such inherited biases can be effectively removed or mitigated *after* distillation with a modified SiD in a fully data-free manner.
>   - Incorporating SFD-style unlearning into SiD-DiT offers a promising direction for reducing teacher-induced artifacts while preserving the benefits of our unified distillation framework.
>
> [1]: Chen, T., Zhang, S., & Zhou, M. Score Forgetting Distillation: A Swift, Data-Free Method for Machine Unlearning in Diffusion Models. In ICLR 2025.

---

### Official Review · Reviewer_8fp7 · 2025-11-01

**Soundness:** 3
**Presentation:** 3
**Contribution:** 2
**Rating:** 6
**Confidence:** 3

**Summary:**

This paper extends Score identity Distillation (SiD) to pretrained text-to-image flow-matching models with DiT backbones. The authors first derive a unified view of Gaussian diffusion and flow matching via Bayes’ rule and conditional expectations, showing that their optimal solutions are identical and differ only in the weight-normalized timestep distribution. They then apply SiD with minimal adjustments to flow matching models, demonstrating 4-step generation in both data-free and data-guided settings without teacher finetuning or architectural changes. Experiments validate robustness across model sizes (0.6B–12B) and noise schedules.

**Strengths:**

1. The derivation is clean, self-contained, and resolves a common misconception: diffusion and flow matching are not fundamentally different under Gaussian assumptions. The reweighting perspective (Eq. 14, Figure 2) is insightful and practically useful.
2. Distilling distinct DiT-based models with a single codebase and hyperparameter set is impressive. Data-free 4-step generation from FLUX.1-DEV (12B) is a significant practical result.
3. Unlike SANA-Sprint (which requires TrigFlow conversion + real data), SiD works “out of the box” on both Rectified Flow and TrigFlow checkpoints. This is a clear win for knowledge transfer.

**Weaknesses:**

1. The core method is SiD, previously validated on U-Net diffusion models. The main claim—“it works on DiT flow models too”—while practically important, is incremental over prior SiD work.
2. Figure 1 is compelling, but no side-by-side with teacher or baselines (e.g., 50-step teacher, SANA-Sprint, DMD).
3. The paper contrasts with SANA-Sprint (consistency distillation) but does not compare to trajectory distillation on the same models.

**Questions:**

1. Can you provide FID/CLIP scores for the teacher (50-step) vs. 4-step SiD vs. SANA-Sprint on the same prompts and seeds?

---

> ### Author Response · Authors · 2025-11-24
>
> We appreciate the reviewer’s recognition of our contribution in providing a unified framework and codebase for distilling diverse DiT models. Below, we address the concerns regarding contribution, empirical comparisons, and methodological sensitivity.
>
> ---
>
> **1. Contribution Beyond Prior Work**
>
> * **Main contribution:**
>   - By establishing both *objective equivalence* and *noise-schedule equivalence* between diffusion and flow matching, **SiD-DiT provides a single, principled distillation pipeline** applicable across a wide range of DiT-based flow models, independent of their original training objective, parameterization, or noise schedule.
>
> * **Breadth of applicability:**
>   - Modern DiT-based flow models vary widely (Efficient-DiT, MMDiT, DC-AE variants; rectified flow; TrigFlow).
>   - Extending a diffusion-based distillation method to such a heterogeneous space requires revisiting **loss weighting**, **timestep parameterization**, and **noise-schedule compatibility** (see Section 2).
>   - **SiD-DiT fills a conceptual and practical gap** by serving as an architecture-agnostic, scheduler-agnostic, and training-agnostic distillation framework.
>
> * **Scalability contribution:**
>
>   - Traditional SiD requires backpropagation through the teacher and fake score networks, making FP32 or mixed-precision necessary in prior T2I work. FP16 degrades performance in T2I models, limiting scalability.
>
>   - **We demonstrate stable SiD-DiT training in pure BF16 under FSDP**, enabling practical distillation of large models such as **SD3.5-Large** on a single node with eight 80GB GPUs.
>
>   - This shows SiD-DiT is not only theoretically unified but also **scales effectively** to modern large-model regimes.
>
> ---
>
> **2. Regarding Weakness 2 (Qualitative Comparison)**
>
> * **Why direct visual comparison is not possible:**
>   Figure 1 reports **SD3.5-Large** results. Methods like **SANA-Sprint** or **DMD2** use *different teacher families*, and corresponding DMD2 checkpoints for SD3/SANA are not available.
>
> * **What we added:**
>   To strengthen within-family comparisons, we regenerated images with **fixed seeds and prompts** for both **SD3.5** and **SANA** teachers.
>   These matched visual comparisons are now included in **Figures 11 and 12** of the appendix.
>
> ---
>
> **3. Regarding Weakness 3 and Question 1**
>
> * **Expanded comparisons:**
>   Revised **Table 1** now includes **multi-step trajectory distillation** results.
>   SiD-DiT consistently provides:
>
>   * superior image quality,
>   * competitive human-preference alignment,
>   * and stable performance across architectures.
>
> * **Conclusion:**
>   These results highlight the **robustness and generality** of SiD-DiT relative to existing fast-distillation approaches such as SANA-Sprint.

---

### Official Review · Reviewer_43LH · 2025-11-01

**Soundness:** 2
**Presentation:** 2
**Contribution:** 1
**Rating:** 2
**Confidence:** 5

**Summary:**

This paper presents a framework for applying score distillation to DiT-based flow-matching models, termed SiD-DiT.
The authors reformulate the score distillation objective using the analytical relation between score functions and velocity fields, enabling distillation directly from flow-matching teachers (e.g., SANA, SD3.5, FLUX).
They employ Fisher divergence minimization for training and further stabilize learning via a fake flow-matching network and an optional adversarial refinement module.
Experiments on text-to-image generation demonstrate that the proposed method achieves reasonable few-step generation quality, with qualitative improvements under different noise schedules.

**Strengths:**

* Provides a unified formulation of score distillation in the flow-matching context, clarifying how SDS-like losses can operate on velocity-based models.
* The training framework (SiD-DiT) is straightforward, practical, and compatible with large-scale DiT backbones.
* Empirical results show that few-step distillation remains feasible for modern flow-matching models, which is of practical interest for efficiency-oriented generative modeling.
* The paper is clearly written, with good visualizations and comparisons of noise schedules that help explain the method’s behavior.

**Weaknesses:**

1. **Limited novelty over existing flow-based distillation methods.**
The use of score distillation on flow-based models has already been explored in several prior works such as CausVid [1], SenseFlow [2], etc. The proposed approach actually does not introduce a new theoretical formulation or algorithmic mechanism, or the paper should more clearly articulate how its method differs conceptually or practically from these existing approaches.

2. **Score–velocity equivalence already established.**
The relationship between score functions and velocity fields has been explicitly derived in Equation (9) of SiT[3], which provides a rigorous mapping between diffusion and flow formulations. It remains unclear whether the ``score target" used in this paper deviates from that known formulation or simply reuses it within a different loss setup.

3. **Training procedure closely resembles DMD but lacks comparison.**
The proposed alternating optimization between a generator and a ``fake score" model, under Fisher divergence minimization, is highly similar to the DMD training pipeline. A fair experimental comparison against DMD/DMD2 based method on flow-matching models would be essential to demonstrate any empirical advantage or stability improvement.

4. **Experimental analysis is incomplete.**
The experiments do not provide sufficient coverage for strong baselines or fair settings:
* Table 1 omits the 4-step result for SANA TrigFlow 1.6B, showing only the 1-step case.
* Table 2 compares only with FLUX-DEV but excludes other diffusion distillation baselines, such as Hyper-FLUX, FLUX-turbo-alpha, etc.
This limited analysis makes it difficult to gauge the relative position and contribution of the proposed method within the broader diffusion-distillation literature.

[1] From Slow Bidirectional to Fast Autoregressive Video Diffusion Models.
[2] SenseFlow: Scaling Distribution Matching for Flow-based Text-to-Image Distillation.
[3] SiT: Exploring Flow and Diffusion-based Generative Models with Scalable Interpolant Transformers.

**Questions:**

Please see the Weaknesses.

---

> ### Author Response · Authors · 2025-11-24
> **Rebuttal to Reviewer 43LH: Request for Clarification on Evaluation and Scoring**
>
> We thank Reviewer 43LH for recognizing the empirical strengths of SiD-DiT, including its robustness across architectures and the clarity of our visualizations and noise-schedule comparisons.
>
> We respectfully observe a potential inconsistency between the review’s qualitative and numerical assessments. While the review describes our pipeline as:
>
> * “straightforward, practical, and compatible,”
> * “clearly written,”
> * offering “good visualizations,” and
> * demonstrating results “of practical interest,”
>
> the numerical ratings for **soundness**, **presentation**, and **contribution** were all marked as *fair* or *poor*.
>
> We kindly request reconsideration and possible recalibration of the scores, as the written feedback appears substantially more positive than the numerical ratings suggest.
>
> Below, we provide our point-by-point response.

---

> > ### Author Response · Authors · 2025-11-24
> >
> > **1. Weakness 1: Limited novelty compared to existing flow-based distillation methods (e.g., CausVid, SenseFlow).**
> >
> > * **Our Response:**
> >
> >   * Novelty assessments are inherently subjective, but key differences exist between SiD-DiT and the cited works.
> >   * **CausVid**:
> >
> >     * Primarily a scaled-up DMD2 applied to *video diffusion*, not modern flow-matching T2I architectures.
> >     * Not directly comparable to large-scale T2I models such as SANA, SD3.5, or FLUX.
> >   * **SenseFlow**:
> >
> >     * A **concurrent** submission to ICLR 2025, with no released code or checkpoints for controlled comparison.
> >     * Requires **real data** for distillation; unlike SiD-DiT, it **cannot** operate in a **data-free** manner and depends on large-scale high-quality real datasets for distillation.
> >     * It can be viewed as a scaled-up version of DMD2 aimed at extending it to larger models. The additional tricks and components it introduces are **complementary rather than competing** with SiD-DiT.
> >     * In principle, one could substitute DMD2 in SenseFlow with **SiD-DiT** and potentially observe further improvements.
> >
> > * **Conceptual and Practical Advances of SiD-DiT:**
> >
> >   1. A **Fisher divergence–based score-distillation objective** enabling fully *data-free* distillation.
> >   2. Integration of a **joint GAN-style refinement component** for enhanced perceptual quality.
> >   3. A **unified time-reweighting scheme** operating across diffusion and flow formulations, including **TrigFlow**, which prior works do not address.
> >   4. Demonstrated applicability across diverse architectures (**MMDiT**, **Efficient-DiT + DC-AE**) and noise schedules (**TrigFlow**), which are not covered by CausVid or SenseFlow.
> >   5. Our empirical results also address longstanding concerns regarding the scalability of SiD. Because SiD requires backpropagation through the teacher, FP32 implementations incur substantial memory overhead. Prior SiD-based T2I models were therefore constrained to FP32 or mixed precision, as FP16 computation, although worked well on ImageNets, was shown to degrade performance in T2I setting and thus limited overall scalability. We show that SiD can be trained stably in pure BF16 under FSDP, enabling practical distillation of large-scale T2I models such as SD3.5-Large on a single node with eight 80GB GPUs. These findings demonstrate that SiD-DiT is not only theoretically unified but also practically viable at modern model scales.
> >
> > * **Summary:**
> >   SiD-DiT extends the conceptual scope and practical applicability of score distillation beyond existing flow-based methods.
> >
> >
> > **2. Weakness 2: Score–velocity relationships have already been derived (e.g., in SiT Eq. 9).**
> >
> > * **Our Response:**
> >
> >   * While SiT presents an epsilon–velocity equivalence, our analysis extends and generalizes this relationship in several important ways:
> >
> >     * **Score-based interpretation:**
> >       * The SiT paper (Ma et al., 2024) is explicitly cited and credited in our work for illustrating the epsilon–velocity relationship. Our contribution **complements and expands** this analysis by approaching the problem purely from the perspective of **loss design and optimal solutions**, without relying on the specifics of reverse diffusion or ODE/SDE formulations.
> >       * Our formulation shows that the key distinctions across objectives—**x-prediction, v-prediction, epsilon-prediction, flow matching, and TrigFlow**—stem primarily from **timestep distributions and reweighting within the training loss**, which influence learning dynamics and empirical behavior despite their shared structural similarities.
> >        * Importantly, these differences matter far less when distilling with **SiD-DiT**: our method applies a **single unified timestep distribution and weighting scheme**, yet still achieves strong performance across distinct DiT-based T2I models.
> >
> >
> >     * **Extension to TrigFlow:**
> >       * In addition to unifying the standard diffusion and flow-matching formulations, **SiD-DiT also unifies TrigFlow**, which has been adopted in recent high-performing text-to-image flow architectures.
> >       * Notably, **SiD-DiT works for both SANA flow-matching and SANA TrigFlow teachers with essentially a single line of code change**, demonstrating the generality of our unified formulation.
> >       * In contrast, SANA-Sprint requires **adapting SANA flow-matching into a TrigFlow framework** before continuous consistency distillation can be applied, underscoring the practicality and generality of our approach.
> >
> > * **Summary:**
> >   Our theoretical framework provides a substantially broader and practically necessary equivalence for modern flow-matching T2I distillation, going beyond the scope of SiT. While SiT focuses primarily on improving the DiT backbone, our formulation is explicitly designed to unify and enable **distillation** across diverse flow-matching objectives.

---

> > > ### Author Response · Authors · 2025-11-24
> > >
> > > **3. Weakness 3: The alternating optimization with a “fake-score’’ model resembles DMD; comparison missing.**
> > >
> > > * **Our Response:**
> > >
> > >   * We kindly ask the reviewer to consider a broader historical perspective when comparing DMD and SiD. The critic–actor (or discriminator–generator) alternating update pattern traces back to **GAN** and **Diffusion-GAN**, and subsequently appears in **Variational Score Distillation**, **Diff-Instruct**, **DMD**, and **SiD**, where the teacher model is incorporated to guide the alternating updates.
> > >
> > >   * It is true that DMD/DMD2 and SiD share a similar high-level alternating structure. However, the **differences between the two families of methods have been extensively documented** in the literature:
> > >     * **Objective differences:**
> > >       *DMD/DMD2* optimize KL-style distribution matching objectives, whereas *SiD* minimizes **Fisher divergence** via score gradients.
> > >     * **Memory vs. performance trade-off:**
> > >       * DMD has lower memory requirements.
> > >       * SiD requires gradients through the teacher and fake score networks (higher memory) but consistently yields **stronger empirical performance** on standard benchmarks.
> > >     * **Data requirements:**
> > >       * DMD/DMD2 often require **real data** to stabilize training.
> > >       * SiD—and SiD-DiT by extension—can operate in a **fully data-free** manner.
> > >
> > >   * **Existing comparisons:**
> > >     * Direct comparisons between DMD/DMD2 and SiD have already been conducted extensively across **CIFAR-10**, **ImageNet**, **Stable Diffusion 1.5**, and **Stable Diffusion XL** in prior work.
> > >     * However, **no publicly available DMD/DMD2 checkpoints or training pipelines exist** for the large-scale DiT-based T2I models considered in our paper (e.g., **SANA**, **SD3.5**), making controlled comparisons difficult at this time.
> > >
> > >   * **Regarding SenseFlow:**
> > >     * SenseFlow (“Scaling Distribution Matching for Flow-based Text-to-Image Distillation”), which reports DMD2-style results, is a **concurrent submission** and does **not release checkpoints or evaluation code**, again preventing apples-to-apples comparison.
> > >     * If the reviewer considers SenseFlow a strong baseline, we appreciate that perspective; importantly, its improvements over DMD2 appear **complementary rather than conflicting** with SiD-DiT. In principle, one could substitute DMD2 with SiD-DiT in such frameworks and potentially obtain further improvements.
> > >
> > >
> > > **4. Weakness 4: Missing comparisons (e.g., SANA TrigFlow 1.6B 4-step) and incomplete evaluation against other baselines.**
> > >
> > > * **Our Response:**
> > >
> > >   * We have expanded the experimental results as follows:
> > >
> > >     * SiD-DiT **outperforms 4-step SANA-Sprint** on both the 0.6B and 1.6B models in FID and CLIP.
> > >     * SiD-DiT is **competitive** with SANA-Sprint in human preference metrics at 0.6B and **surpasses** it at 1.6B.
> > >     * Added results for **HyperFLUX** and **FLUX-Turbo-Alpha** to broaden comparison with state-of-the-art fast flow models.
> > >
> > > * **Summary:**
> > >   Thank you for the suggestion. The expanded experiments now provide a clearer and more comprehensive view of SiD-DiT’s position among modern efficient T2I generators.

---

> ### Comment · Reviewer_43LH · 2025-11-25
> **Official Comment**
>
> Thank you for providing the detailed rebuttal. I have read the responses carefully, along with the other reviewers’ comments and the authors’ clarifications, which addressed some of my earlier concerns and helped resolve parts of my confusion regarding the contributions of SiD-DiT. I still believe that the performance comparisons in the initial submission were somewhat risky, as the setup might give readers an impression that is not entirely clear or fully fair. While I remain inclined to view SiD-DiT primarily as an extension of prior work (SiD, SiDA, SiD-alpha) rather than a fundamentally novel methodological contribution, I acknowledge that the approach demonstrates strong training stability across different flow matching backbones and noise schedules. Given that the paper is titled *“Score Distillation …”* and that DMD represents a major and influential line of work in score distillation, it is natural for readers to wonder about the advantages of SiD-DiT relative to DMD.
>
> In addition, I am curious about the training cost of SiD-DiT, and I would appreciate it if the authors could report resource usage (e.g., GPU hours) in the Table. 4/5 of the revised version.
>
> Based on the authors’ responses and the clarified contributions, I will reconsider my score accordingly.

---

> > ### Author Response · Authors · 2025-11-26
> > **Addressing Reviewer Concerns with Added Figure 4 and Table 6 on SiD-DiT Convergence and Cost**
> >
> > Thank you very much for taking the time not only to read our rebuttal carefully, but also to review the other discussions in detail. We are glad that our responses were able to resolve several of your earlier concerns. Your points were valid and fair, and we agree that the initial presentation could have been clearer about how SiD-DiT compares with DMD/DMD2 and other prior score-distillation approaches.
> >
> > To directly address your remaining concerns, particularly regarding training cost, we have added Figure 4 to the main paper, which illustrates how rapidly SiD-DiT converges across all 9 evaluated DiT models. In addition, we have included Table 6 in the Appendix to report the estimated training times for both reaching near-converged metrics and training to the final checkpoint.
> >
> > We would also like to emphasize that SiD-DiT reaches to good performance very quickly. For all models except FLUX, approximately 100k images, corresponding to roughly 400 iterations at a batch size of 256, are sufficient to reach near teacher performance. This translates to about half an hour of training for SANA, a few hours for SANA (TrigFlow), SD3-Medium, and SD3.5-Medium, and less than a day for SD3.5-Large.
> >
> > We appreciate your thoughtful feedback and your willingness to reconsider your score in light of the clarified contributions and additional results. Please let us know if any further details would be helpful.

---

### Author Response · Authors · 2025-12-04
**Final Response to Reviewer Feedback and Summary of Revisions**

Dear AC,

Through our detailed rebuttal and revised manuscript, we believe we have fully addressed all questions and concerns. Below is a brief summary of the key reviewer comments and the substantive revisions we have made in response.

We find that the reviewers’ questions mainly centered on (i) in-depth comparison with additional baselines, and (ii) the contribution of SiD-DiT beyond prior work:

- **New baselines:** In the revision, we have added all reproducible, open-source baselines mentioned by the reviewers, enabling a more thorough and fair empirical comparison.

- **Technical contribution:** We have also clarified the core contribution of SiD-DiT: it derives a unified score-distillation framework that in theory applies to any flow-matching objective and noise schedule, rather than being tied to a specific formulation. The resulting unified distillation pipeline for flow-based T2I models now covers almost all open-source flow-based T2I backbones and achieves competitive performance across the board, a direction that has received consistently positive feedback from the reviewers.

In the revised manuscript, we have substantially strengthened the paper with a comprehensive set of new experiments and clarifications:

 - **Additional quantitative metrics for image generation:** We now include more baselines (SANA-Sprint, SANA-Sprint teacher, HyperFlux, and FLUX-Schnell) against the corresponding SiD-DiT models, and we report detailed human preference scores for all SiD-DiT variants and baselines.

- **Detailed analysis of training cost:** We provide a detailed analysis of training cost, reporting estimated GPU hours and the number of k-images processed for all checkpoints in the manuscript. Because SiD-DiT achieves strong performance early in training, reaching competitive quality well before full convergence, we also report the GPU hours needed to reach this early high-quality regime, underscoring not only its state-of-the-art final performance but also its training efficiency.

 - **Additional qualitative comparisons:** We add more visual examples comparing SiD-DiT to baselines with prompts in Figure 1 and random seeds fixed for a fair, side-by-side evaluation.

Collectively, these additions address nearly all concerns regarding the performance and training cost of the SiD-DiT pipeline, and Reviewer 43LH has explicitly acknowledged our efforts. Although we have so far received follow-up feedback only from Reviewer 43LH, their updated comments are noticeably more positive than their initial assessment. We believe that the new illustrations and detailed explanations in the revision directly and comprehensively resolve the questions they raised.

We hope these substantial additions and clarifications fully resolve the reviewers’ concerns, and we would be grateful if you could take them into account in your evaluation.

Thank you again for your time,

Authors of SiD-DiT

---

### Meta-Review · Area_Chair_AeuC · 2026-01-08

**Summary:**

This paper introduces SiD-DiT, extending Score Identity Distillation (SiD) to DiT-based flow-matching text-to-image models. The authors provide a clean unification showing that diffusion and flow-matching objectives share the same optimum under Gaussian assumptions, differing mainly by timestep reweighting. Building on this view, they apply SiD to pretrained flow-based T2I models (SANA, SD3/3.5, FLUX), enabling few-step (typically 4-step) generation without teacher finetuning or architectural modification. The empirical results demonstrate that SiD-DiT is stable across architectures, noise schedules (including TrigFlow), and model scales, including very large models, which is technically impressive.

**Reviewer Concerns:**

Several concerns raised in the initial reviews were meaningfully addressed during rebuttal: the authors expanded experimental coverage, added training-cost and convergence analyses, improved within-family comparisons (e.g., against SANA-Sprint), and clarified implementation details. These revisions strengthen the empirical credibility of the work and show that the proposed pipeline is robust and practically deployable.

However, key concerns remain. Across reviewers, there is strong consensus that the contribution is primarily incremental rather than conceptual. The core methodology is SiD, and the main claim—that SiD can be applied effectively to DiT-based flow-matching models—while practically useful, is viewed as an extension of prior SiD/SiDA-style work rather than a fundamentally new distillation principle. The theoretical equivalence between diffusion and flow matching is known in the literature; although the presentation here is clean and unified, it is seen as a refinement rather than a new insight. The work remains best characterized as a strong, carefully engineered extension of SiD to a broader model class, rather than a new methodological advance.

Additionally, despite added discussion, the paper does not fully resolve concerns about positioning relative to other major distillation paradigms, especially DMD-style methods. While direct large-scale comparisons are difficult, reviewers consistently felt that the advantages of SiD-DiT over these alternatives remain insufficiently crystallized. The weaker performance on FLUX models also raises unresolved questions about stability versus architecture-specific limitations, which are acknowledged but not fully analyzed.

**Reviewer Scores:**

Revisions improved clarity and empirical support, and some reviewers indicated a willingness to reconsider their scores. Nevertheless, the dominant remaining concern—limited novelty relative to prior SiD and distillation literature—persists across multiple reviews and is unlikely to fully dissipate without a stronger conceptual or methodological advance.

---

### Decision · Program_Chairs · 2026-01-26

Reject